# Single cell analyses identify a highly regenerative and homogenous human CD34+ hematopoietic stem cell population

Fernando Anjos-Afonso [1,8✉], Florian Buettner[2,3,4], Syed A. Mian [5], Hefin Rhys [6], Jimena Perez-Lloret[7], Manuel Garcia-Albornoz [5], Namrata Rastogi[1], Linda Ariza-McNaughton[5] & Dominique Bonnet [5,8✉]

The heterogeneous nature of human CD34[+] hematopoietic stem cells (HSCs) has hampered our understanding of the cellular and molecular trajectories that HSCs navigate during lineage commitment. Using various platforms including single cell RNA-sequencing and extensive xenotransplantation, we have uncovered an uncharacterized human CD34[+] HSC population. These CD34[+]EPCR[+](CD38/CD45RA)[−] (simply as EPCR[+]) HSCs have a high repopulating and self-renewal abilities, reaching a stem cell frequency of ~1 in 3 cells, the highest described to date. Their unique transcriptomic wiring in which many gene modules associated with differentiated cell lineages confers their multilineage lineage output both in vivo and in vitro. At the single cell level, EPCR[+] HSCs are the most transcriptomically and functionally homogenous human HSC population defined to date and can also be easily identified in post-natal tissues. Therefore, this EPCR[+] population not only offers a high human HSC resolution but also a well-structured human hematopoietic hierarchical organization at the most primitive level.

[1] Haematopoietic Signalling Group, European Cancer Stem Cell Institute, School of Biosciences, Cardiff University, Cardiff, UK. [2] German Cancer Research Center (DKFZ), Heidelberg, Germany. [3] German Cancer Consortium, Heidelberg, Germany. [4] Frankfurt University, Frankfurt, Germany. [5] Haematopoietic Stem Cell Lab, The Francis Crick Institute, London, UK. [6] Flow Cytometry Facility, The Francis Crick Institute, London, UK. [7] Advance Sequencing Facility, The Francis Crick Institute, London, UK. [8] These authors contributed equally: Fernando Anjos-Afonso, Dominique Bonnet. ✉email: dosanjosafonsof@cardiff.ac.uk; dominique.bonnet@crick.ac.uk

Blood is one of the most highly regenerative tissues, with approximately $10^{11}$–$10^{12}$ cells being produced daily in adult human bone marrow. Continuous blood supply throughout life relies on a rare population of hematopoietic stem cell (HSC). Most of our understanding of HSC hierarchical organization comes from the murine studies. Indeed, the ability to prospectively characterize murine HSCs has enabled us to draw a detailed cellular network of the mouse bone marrow, however, this advanced resolution of human HSCs still lags far behind. Although a very rare HSC population that is negative for CD34 expression has recently been demonstrated to be hierarchically above the CD34$^+$ cells[1–3], human hematopoiesis is considered to be mostly sustained by CD34$^+$ hematopoietic stem/progenitor cells (HSPCs). Within the primitive CD34$^+$ HSPCs, phenotypically defined as CD34$^+$CD38$^-$CD45RA$^-$, Notta et al. further isolated four sub-populations based on CD90 (a GPI-linked glycoprotein) and CD49f (integrin α6) expression (simply as CD90$^{+or-}$CD49f$^{+or-}$)[4]. CD90$^+$CD49f$^+$ and CD90$^-$CD49f$^-$ fractions were shown to be highly enriched in HSCs and MPPs (multipotent progenitors) respectively[4–6]. Since their description, they have been widely used as main stem and early progenitor reference populations. However, the CD90$^+$CD49f$^+$ population remains very heterogeneous[7] with a suboptimal purity of 5 to 10%, which prohibits studies of a pure HSC population. This is due, in part, to the difficulties in separating the negative and positive populations, as CD49f is weakly expressed on human CD34$^+$ HSPCs. This has hindered our ability to further investigate the hierarchical relationships that may exist between these HSPC populations and also map their molecular profile that govern cell-fate decisions[6–12].

While dissecting the potential relationship between these four populations, we uncovered that not only the CD90$^+$CD49f$^+$ but also the CD90$^-$CD49f$^+$ population contains self-renewing HSCs. More importantly, these two CD49f$^+$ populations are inter-convertible and give rise to all CD34$^+$ HSPCs. Using single-cell RNA sequencing (scRNA-seq), we found EPCR (endothelial protein C receptor) as a common cell surface antigen that captures most, if not all, the self-renewing repopulating cells in the two CD49f$^+$ populations. Based on further scRNA-seq, comprehensive in vivo testing, and molecular characterization, we revealed that these CD34$^+$EPCR$^+$(CD38/CD45RA)$^-$ (simply as EPCR$^+$) HSCs have a high repopulation and self-renewal abilities and an in vivo balanced-myeloid prone differentiation capacity that is supported by a unique transcriptomic wiring. Importantly, this work also supports the hypothesis that EPCR$^+$ cells represent the purest human HSC population described to date.

## Results

**Interconvertible CD90$^+$49f$^+$ and CD90$^-$49f$^+$ populations contain bona fide HSCs.** We began investigating the relationship between the four cell sub-fractions (CD90$^+$CD49f$^+$, CD90$^-$CD49f$^+$, CD90$^+$CD49f$^-$, and CD90$^-$CD49f$^-$) by determining their repopulating capacity and frequencies (commonly denoted as SCID-repopulating cell or SRC) in NSG (non-obese diabetic/severe combined immunodeficiency, gamma chain null) mice using stringent gating, double flow-sorting and purity-checked protocols (Supplementary Fig. 1a–c). Both CD49f$^+$ populations had the highest SRC frequencies (1 in 8.37 for CD90$^+$CD49f$^+$ and 1 in 17.5 for CD90$^-$CD49f$^+$ cells; Fig. 1a and Supplementary Data 1), while the other two CD49f$^-$ populations had the lowest SRC frequencies (Fig. 1a). These results are similar to the previous reports that used intra-bone transplantation assays[4], thus supporting the robustness of our optimized intra-venous (i.v) xenotransplantation method (see "Methods"). To induce a higher regenerative demand, we used a transplantation

protocol comprising of $12 + 12$ weeks (wks) for 1$^{ry}$ and 2$^{ry}$ transplants respectively with a limited cell dose of 1 SRC/mouse, instead of using a large cell dose and evaluating engraftment at a longer time point. When transplanting ~1 SRC/mouse, we observed a decrease in human engraftment derived from CD90$^+$CD49f$^-$ and CD90$^-$CD49f$^-$ cells from 5 to 12 wks, that eventually dropped off (<0.01%) at 18–24 wks post-transplant[4] (not data shown). However, higher persistent engraftment was observed for the two CD49f$^+$ populations (Fig. 1b). Importantly, when CD34$^+$CD38$^{lo/-}$ HSPCs derived from either of the two CD49f$^+$ populations were re-transplanted at limited cell dose (600 CD45$^+$CD34$^+$CD38$^{lo/-}$ cells) it generated successful secondary grafts (Fig. 1b). Thus, our data clearly demonstrates that not only the CD90$^+$CD49f$^+$ but also CD90$^-$CD49f$^+$ populations have self-renewal capacity. Additionally, only the two CD49f$^+$ cell populations generated substantial CD34$^+$CD38$^{lo/-}$ HSPCs in vivo (Fig. 1c), comprising up to ~20% of the human graft at 5 wks after transplanting just ~1 SRC (Supplementary Fig. 1d). Detailed phenotypic analyses within the human HSPCs highlighted that both CD49f$^+$ populations not only self-renewed, but also gave rise to the entire CD34$^+$ compartment (Fig.1d and Supplementary Fig. 1f). In contrast, mice transplanted with CD90$^+$CD49f$^-$ population had expanded and produced MPPs which in turn gave rise to subsequent downstream progenitors (CD45RA$^+$), therefore indicating a sequential hierarchical organization. Importantly, we uncovered that CD90$^-$CD49f$^+$ cells produced CD90$^+$CD49f$^+$ cells in vivo and vice-versa (Supplementary Fig. 1f and Fig. 1d) hence, demonstrating that both CD49f$^+$ sub-fractions are interchangeable and lie at the top of the CD34$^+$ stem cell hierarchy (Supplementary Fig. 1g).

**Transcriptomic and in vivo studies identify an undescribed human HSC population.** Next, we hypothesized that a common sub-population of cells may exist in the CD90$^+$CD49f$^+$ and CD90$^-$CD49f$^+$ populations that generated similar in vivo outputs. In order to determine candidate gene markers that could be useful to identify this hypothetical population, we performed single cell RNA-sequencing (scRNA-seq) in the two CD49f$^+$ populations (469 cells; denoted as P1 and P2 for CD90$^+$CD49f$^+$ and CD90$^-$CD49f$^+$ cells, respectively). We used a diffusion map instead of t-SNE or UMAP approach to visualize the single-cell data since this algorithm is designed to capture smooth transitions between cells. Interestingly, our single cell analysis revealed a continuum of cells between those that are enriched in CD90$^+$CD49f$^+$ (P1; Fig. 2a left) and those that are enriched in CD90$^-$CD49f$^+$ (P2; Fig. 2a right). We also performed bulk RNA-seq in the four populations (CD90$^+$CD49f$^+$ (P1), CD90$^-$CD49$^+$ (P2), CD90$^+$CD49f$^-$ (P3) CD90$^-$CD49f$^-$ (P4) (Fig. 2b). In this analysis we obtained a differential expression (DE) list of genes (list C) which is a set of potential marker genes that was generated as the union of genes that were DE between the two CD49f$^+$ versus (vs) the two CD49f$^-$ populations: P1 vs (P3 + P4), list A and P2 vs (P3 + P4); list B. This list C was then used to guide and infer the internal cell state of the single cells, which was augmented with the highly variable genes from the scRNA-seq data itself, thus allowing to quantify the transition between P1 and P2 cells. We then quantified this transition using the concept of diffusion time CITE and identified three sub-populations (SP1, SP2, and SP3) by dividing the diffusion time into terciles. The three sub-populations correspond to cells falling into the respective terciles: SP1 consisted of 157 cells, with 28.02% of cells in SP1 being CD90$^+$CD49f$^+$ cell; SP3 consisted of 156 cells with 23.07% of cells in SP3 being CD90$^-$CD49f$^+$ cells and SP2 was a common and balanced sub-population of 156 cells consisting of 53.84% CD90$^+$CD49f$^+$ and 46.16% CD90$^-$CD49f$^+$ cells (Fig. 2c)

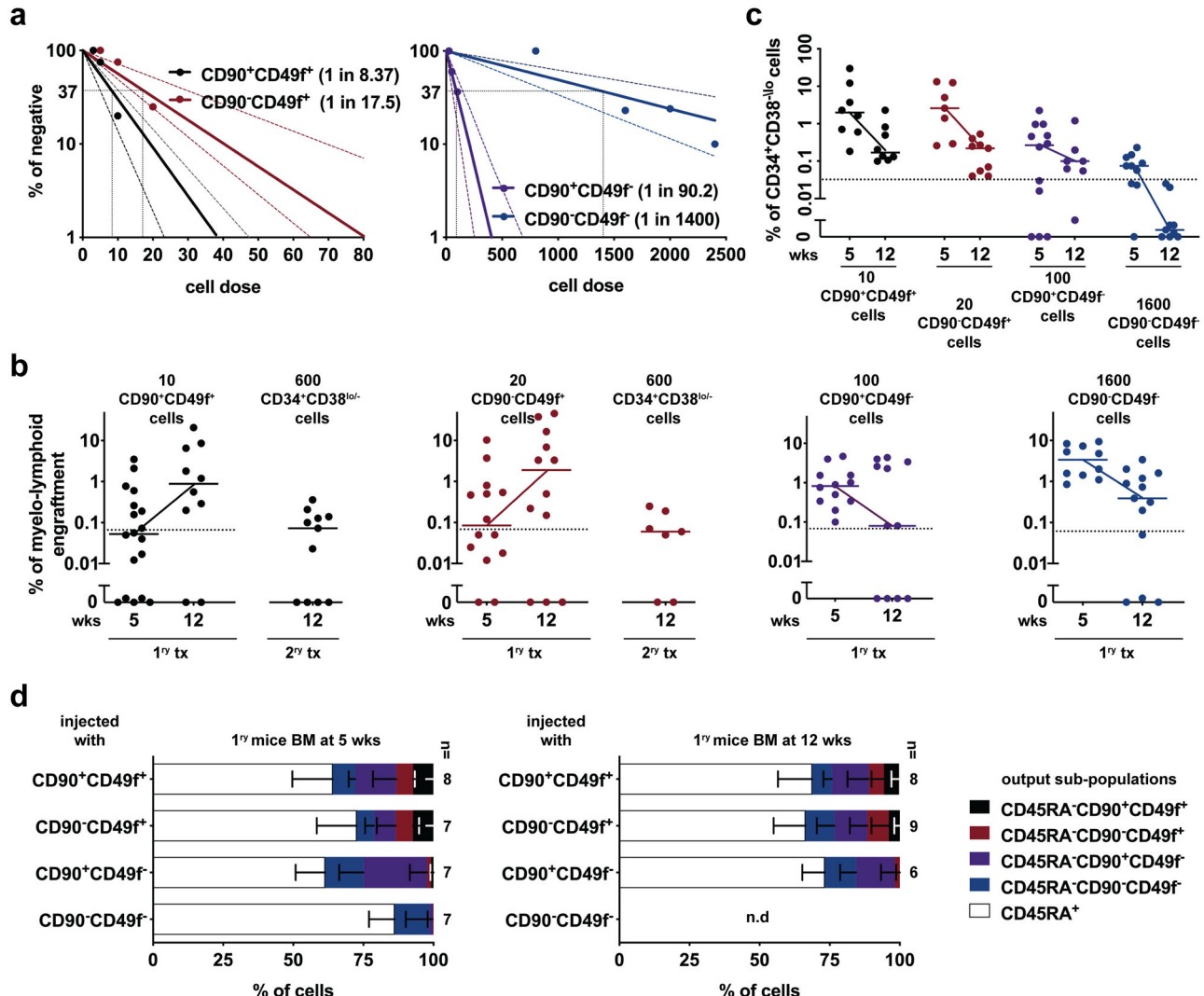

**Fig. 1 Both CD90+49f+ and CD90−49f+ self-renewing populations are interconvertible and give rise to the whole CD34+ compartment. a** SRC frequency within the four most primitive populations measured by limiting dilution assay (LDA) at 12 wks post intravenous (i.v) transplantation. 95% confidence interval is shown (see Supplementary Data 1 for details). To obtain unbiased frequencies, similar number of male and female mice were used (Supplementary Fig. 1e). **b** Cell dose equivalent of ~1 SRC of each population was i.v injected into separate primary NSG recipients (1$^{ry}$ tx) and human graft was determined at 5 and 12 weeks (wks) later ($n = 7$–18 mice). Secondary transplantation (2$^{ry}$ tx) was performed with ~1 secondary SRC dose of CD34+CD38$^{lo/−}$ HSPCs (600 CD45+CD34+CD38$^{lo/−}$ cells) derived from CD90+CD49f+ and CD90−CD49f+ populations. Mice with sufficient human graft (above the dotted line, see "Methods") were used for analysis in (**c**). Each dot represents a mouse ($n = 7$–12 mice); median lines are shown; results were from five independent experiments. **c** Frequency of CD34+CD38$^{lo/−}$ HSPCs within human graft found in the BM of 1$^{ry}$ recipients. Mice with sufficient human graft (above the dotted line) were used for analysis in (**d**); median lines are shown ($n = 7$–12 mice). **d** Frequency of the denoted output sub-populations within CD34+CD38$^{lo/−}$ HSPCs found in mice engrafted at 5 and 12 wks with the indicated cell populations ($n = 6$–9 mice). Percentages were calculated based on the gates shown in Supplementary Fig. 1f. n.d not detected. Bars are the mean values and error bars are the S.D for the no. of mice analyzed. Source data are provided as a Source Data file.

To characterize this common sub-population, we performed a DE analysis, contrasting SP1 vs SP2 and SP3 vs SP2 respectively, using the Wilcoxon rank test. This analysis revealed a set of 950 genes that were DE in both comparisons (Fig. 2d). From the DE genes found in the scRNA-seq dataset (Supplementary Data 3), and the DE genes from the bulk analysis (Supplementary Data 2), we identified *PROCR* (or CD201) as one of the top DE genes in the common sub-set of genes (Fig. 2d). The identification of *PROCR* as one of the DE genes in the bulk analysis (Fig. 2d and Supplementary Data 2) reinforced the results obtained from the scRNA-seq analysis.

As *PROCR* encodes for EPCR (endothelial protein C receptor) and has been recently shown to be expressed on human HSPCs

expanded with the pyrimido-indole derivative UM171[13,14], we sought to further investigate if EPCR can be used to identify and isolate un-cultured human HSCs. First, we verified higher EPCR expression on both CD49f+ compared to the other two CD49f− populations (Supplementary Fig. 2a, b) and also detected a clear small EPCR+ sub-population (Fig. 3a). Interestingly, most of EPCR+ cells were CD38$^{lo/−}$, CD45RA− and most importantly CD49f+ (Fig. 3a, b). To test our hypothesis that EPCR+ cells contained the majority of cells with robust repopulating capacity within the two CD49f+ fractions, we purified and transplanted EPCR− and EPCR+ cells from CD90+CD49f+ and CD90−CD49f+ cells, respectively. Our results clearly show that CD90$^{+or−}$CD49f+ EPCR+ cells robustly repopulated NSG mice with very low cell

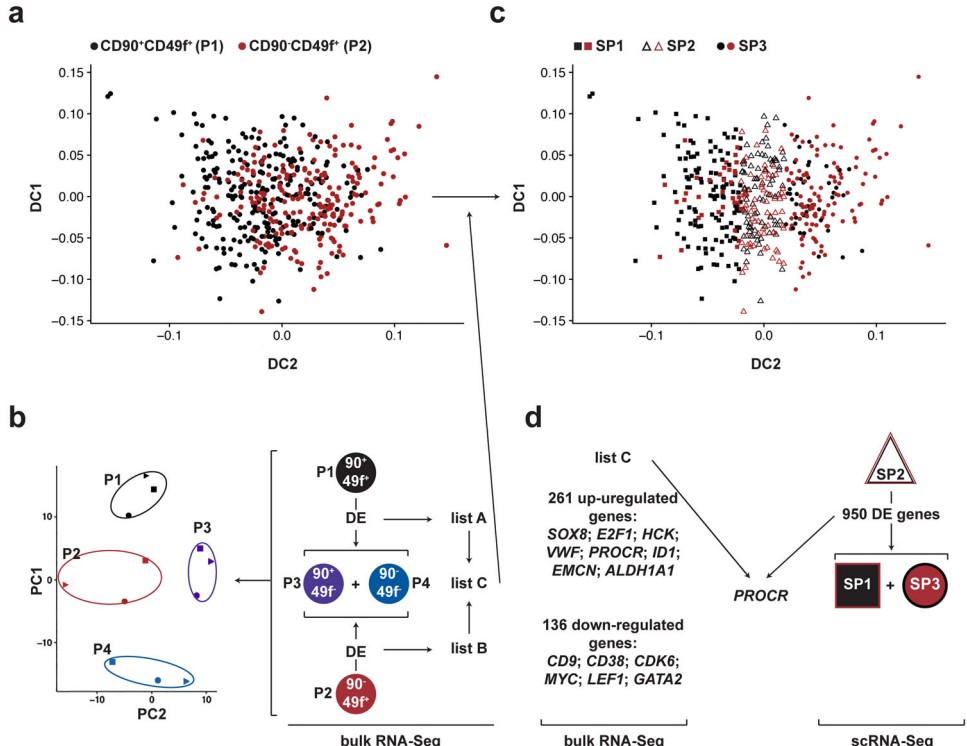

**Fig. 2 Transcriptomic studies identify EPCR as a potential selection marker for the repopulating cells within both CD90⁺CD49f⁺ and CD90⁻CD49f⁺ populations. a** Diffusion map representation of the single cell data generated from 469 QC-passed single CD90⁺CD49f⁺ (P1) and CD90⁻CD49f⁺ (P2) cells. **b** Two-dimensional representation of the bulk RNA-seq performed in the four populations (*n* = 3) by principle component analysis (PCA). DE analyses were performed with the indicated populations: CD90⁺CD49f⁺ (P1) vs ((CD90⁺CD49f⁻ (P3) + CD90⁻CD49f⁻ (P4)) and between CD90⁻CD49f⁺ (P2) vs ((CD90⁺CD49f⁻ (P3) + CD90⁻CD49f⁻ (P4)) and we refer to the respective gene sets as lists A and B. **c** Diffusion map representation of the single cell (*n* = 469) data shown in **a** after the inferred internal cell state was guided by the total RNA-seq data via list C resulting in 3 sub-populations (SP1, SP2, and SP3). **d** Workflow analysis of the total and scRNA-seq data (Supplementary Data 2 and 3 for details). Source data are provided as a Source Data file.

numbers, whereas CD90⁺ᵒʳ⁻CD49f⁺EPCR⁻ cells mostly failed to engraft even with up to 4 times higher cell doses (Fig. 3c). Furthermore, the SRC frequencies in both CD49f⁺ populations increased when EPCR was used as an additional selection marker, which resulted in similar SRC frequencies (Fig. 3d). Therefore, our data clearly demonstrates that the EPCR further delineates the repopulating cells within the two CD49f⁺ populations.

**Highly regenerative EPCR⁺ HSCs sits at the apex of a re-structured CD34⁺ HSPC hierarchy.** We then hypothesized that by using a simple gating strategy (CD34⁺CD38⁻CD45RA⁻EPCR⁺, thereafter as EPCR⁺) we can purify most, if not all, human CD34⁺ HSCs (Fig. 4a). To test this, we began determining the SRC frequency of EPCR⁺ population and obtained a similar and even slightly higher frequency (1 in 3.1 cells; Figs. 4a and 1a). This can be reasoned by the fact that ~25% of the cells that were omitted in between CD90⁺ and CD90⁻ expression in the previous stringent gating strategies were then included. Of note, for these and subsequent experiments we still included CD90 staining in order to sort for CD90⁺EPCR⁻(CD34⁺CD38⁻CD45RA⁻), to directly compare with EPCR⁺ cells (Fig. 4a). In addition to this, since two unknown populations have been identified, we also included MPPs (CD90⁻EPCR⁻(CD34⁺CD38⁻CD45RA⁻)) in some of the in vivo experiments as our reference population in order to link the above data. The SRC frequency of CD90⁺EPCR⁻ population reached a frequency (1 in 119 cells; Fig. 4a) similar to the CD90⁺CD49f⁻ population but ~40-fold lower as compared to the EPCR⁺ cells.

We then i.v transplanted a cell dose close to ~1 SRC (4 EPCR⁺ cells) and observed similar engraftment kinetics as the two CD49f⁺ populations (Figs. 4b and 1b), whereas the engraftment derived from CD90⁺EPCR⁻ cells decreased overtime. The reduced repopulating capacity of the latter population was further supported by the limited CD34⁺CD38⁻ HSPC output in the 1ʳʸ mice (Supplementary Fig. 3a). Importantly, CD90⁺EPCR⁻ cells did not generate EPCR⁺ cells, but only MPPs and MLPs/LMPPs (Fig. 4c and Supplementary Fig. 3b). Based on the similar results obtained (e.g., SRC frequency, kinetics, and progenitor outputs) between CD90⁺EPCR⁻ and CD90⁺CD49f⁻ cells, we assumed that the in vivo outcomes of the refined MPP would be analogous to the MPP population previously defined (CD90⁻CD49f⁻). Indeed, when we transplanted 8000 CD90⁻EPCR⁻ MPPs (~5 CD90⁻CD49f⁻ SRCs) we detected very limited CD34⁺CD38⁻ HSPC output (Supplementary Fig. 3a), and these progenitors mainly produced CD45RA⁺ MLPs/LMPPs in vivo (Fig. 4c), thus demonstrating that they have a limited self-renewal capacity. Importantly, secondary transplantations using limiting HSPC cell dose derived from EPCR⁺ cells resulted in robust secondary engraftment with ~25× higher levels compared to the two CD49f⁺ populations (Figs. 4b and 1b), thus demonstrating that EPCR⁺ cells have the highest self-renewal capacity described to date (Fig. 4c).

We then went on to evaluate whether this EPCR⁺ population could also be detected in adult bone marrow (BM). We found a phenotypically similar cell fraction comprising ~0.03% of CD34⁺ HSPCs (Fig. 4d), which we estimated to be around 1 cell in

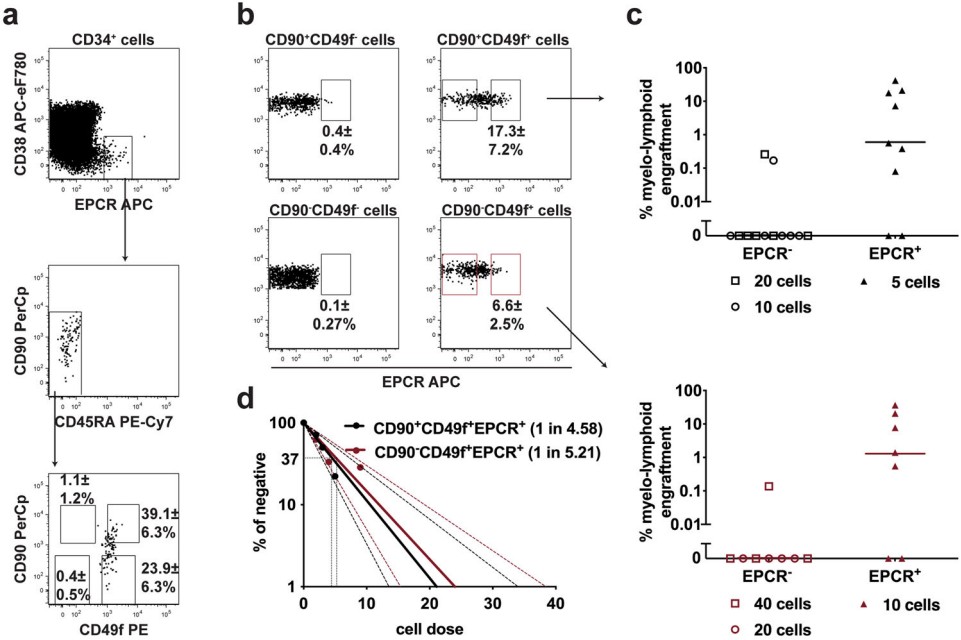

**Fig. 3 In vivo studies support EPCR as a marker for the repopulating cells within both CD90+CD49f+ and CD90−CD49f+ populations. a** Representative phenotypic analysis of EPCR+ cells in cord blood CD34+ cells. Frequencies were calculated based on the gates shown (n = 8 mice) Of note, three different commercial anti-EPCR antibodies gave very similar staining profiles (Supplementary Fig. 2c). **b** Frequency of EPCR+ cells within CD90+CD49f+, CD90−CD49f+, CD90+CD49f− and CD90−CD49f− populations respectively (n = 8 mice). **c** Engraftment of EPCR+ and EPCR− cells within CD90+CD49f+ and CD90−CD49f+ populations (flow-sorted as shown in (**b**)) assayed in NSG mice 12 wks post-transplant with the denoted cell dose. Results were from three independent experiments (n = 7–12 mice); median lines are shown. **d** SRC frequency within (CD34+CD38−CD45RA−) CD90+CD49f+EPCR+ and CD90−CD49f+EPCR+ populations; 95% confidence interval is shown. ± shown is the S.D for the no. of experiments performed. The anti-CD201 clone RCR-227 was used in all these experiments. Source data are provided as a Source Data file.

166,650 to 333,330 mononuclear cells. We next tested the in vivo engraftment potential of these adult BM cells and found indeed that mice injected with as few as 25 BM EPCR+ cells engrafted (Fig. 4e), whereas human engraftment was only observed when 6 times more CD90+EPCR− cells were transplanted. As CD49f expression is almost undetectable in adult BM CD34+ HSPCs (Supplementary Fig. 2d), EPCR provides a better alternative marker to purify BM HSCs. Altogether, our data demonstrate that this EPCR+ population defines the most primitive post-natal CD34+ HSPCs that also has the potential to generate the CD90+EPCR− cells downstream of EPCR+ cells but above the MPPs.

**EPCR+ cells are slow cycling cells with low metabolic features**. To gain insight into the molecular features that govern the functional properties of these cell populations (e.g., self-renewal, engraftment kinetics, etc), we performed bulk RNA-seq in EPCR+ cells and the immediate downstream CD90+EPCR- and MPP progeny fractions. This analysis revealed 547 and 2154 significant DE genes between EPCR+ vs CD90+EPCR− and vs MPP populations respectively (Fig. 5a and Supplementary Data 4), where higher expression of HSC-associated genes such as *ACE*, *ID1*, *ID3*, *HLF*, *HCK*, and *SOX8*[4,12] was found in EPCR+ cells. In contrast, both CD90+EPCR− and MPP cells were enriched in cell cycle/cell activation-associated genes like *MYB*, *MYC*, *CD69*, and *CDK6*. Interestingly, the significance of the differential expression of these genes increased from CD90+EPCR− to MPP, suggesting a further loss of self-renewal and increase in cell activation when primitive cells become MPPs.

Furthermore, we used the entire WikiPathways database to search for biological pathways that could be enriched in each population, and we performed pathway analysis using CAMERA, a competitive gene set test accounting for inter-gene correlation[15].

This analysis revealed that pathways associated with cell cycle/DNA replication, mitochondrial metabolism, and cytoplasmic ribosomal were more enriched in the two progenitor populations as compared to EPCR+ cells (Fig. 4b). This transcription data was then independently confirmed using functional assays. Indeed, EPCR+ cells had significantly divided less in serum-free cultures compared to the other two progenitors with an average of 2.6 as compared to 3.17 (CD90+EPCR−) and 3.44 (MPP) cell divisions in 4 days (Fig. 5c). Mitochondrial membrane potential and ATP production were also significantly lower (1.48- to 1.6-fold and 1.33- to 1.55-fold, respectively) in EPCR+ cells compared to the other two progenitor populations (Fig. 5d, e). Newly translated proteins were determined by measuring the incorporation of OP-Puro (O-propargyl-puromycin), and this was generally low in all the three primitive populations analyzed. However, EPCR+ cells had the lowest while MPPs had the highest protein synthesis capacity (Fig. 5f). Altogether, our data supports that EPCR+ cells have the lowest cycling and metabolic activity features as compared to any CD34+ HSPC population described[10,16,17].

**Differential transcriptomic wiring supports the unique in vivo balanced-myeloid prone differentiation capacity of EPCR+ cells**. We observed a unique feature where primitive EPCR+ cells contributed to a more balanced-myeloid prone differentiation output in vivo (with an ~65 to 35% of CD33+ to CD19+ output ratio) whereas CD90+EPCR− cells and MPPs provided a CD19+-bias output (~25 to 75% and ~10 to 90%, respectively) that is normally observed in NSG mice, like for all the human CD34+ HSPC populations that have been evaluated so far[4,6] (Fig. 6a). We then hypothesized that some level of lineage-priming programs might already be inherently present in primitive EPCR+ cells. To test this hypothesis, we used a set of published gene modules that

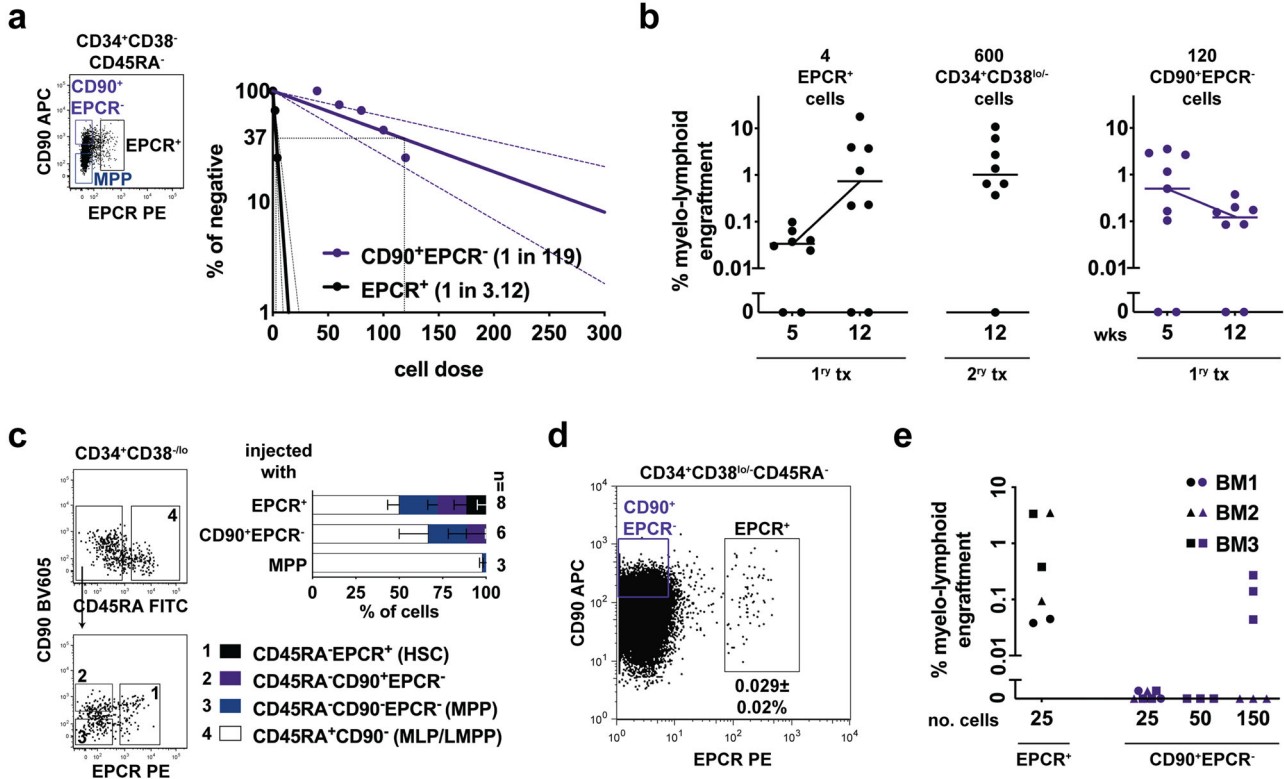

**Fig. 4 Most self-renewing EPCR+ cells are at the top of a newly defined human CD34+ HSPC hierarchy. a** Representative FCM plot illustrating the gating strategy used to sort CB EPCR+, CD90+EPCR- and MPP populations and SRC frequency of EPCR+ and CD90+EPCR- populations; 95% confidence interval is shown. **b** Engraftment kinetics of the indicated cell populations. Mice were transplanted with ~1 SRC dose for 1ry tx and 2ry tx respectively. Results were from three independent experiments ($n = 8–9$ mice); median lines are shown. **c** Representative FCM plots illustrating the phenotype of the denoted sub-populations within CD34+CD38lo/− HSPCs detected at 12 wks in a primary mouse bone marrow transplanted with EPCR+ cells. Cumulative data from all the mice that yielded sufficient HSPCs is shown ($n = 3–8$ mice). The percentages were calculated based on the gates shown (for BMs engrafted with CD90+EPCR− cells and MPPs see Supplementary Fig. 3b). **d** Representative FCM plot showing the presence of EPCR+ and CD90+EPCR− cells in adult BM ($n = 7$). **e** BM derived EPCR+ and CD90+EPCR− cells were i.v injected into primary NSG recipients with the denoted cell dose ($n = 3–7$ mice) and human graft was determined 14–16 wks later. Bars are the mean values and error bars are the S.D for the no. of experiments performed. In the LDA experiments the CD201 clone RCR-227 and the CD201 clone RCR-401 antibodies were used, and in the remaining experiments only the latter antibody was used. Source data are provided as a Source Data file.

is associated with early hematopoietic lineage-priming[8] and found that HSC-related modules were enriched in EPCR+ cells, particularly when compared to MPPs (Fig. 6b). Importantly, EPCR+ cells were found to be enriched in multilineage modules such as the HSC/lymphoid/myeloid *BTG2/HOXA6* and the neutrophil/monocyte/B lymphoid *RERE/BCL6* gene sets and also the neutrophil *CEBPA/CEBPD* gene cluster (Fig. 6b). This supports our observations that EPCR+ cells have similar in vivo lineage outputs as the highest self-renewing mouse HSCs that have been previously characterized of having a balanced with a slight myeloid prone differentiation capacity[18,19]. We then performed additional enrichment analysis using other published gene sets[9] (derived from more differentiated hematopoietic cells) with the aim to find support for the observed differentiation output shift between EPCR+ vs CD90+EPCR−/MPP cells. Indeed, this analysis revealed that the myeloid gene clusters (CD14+ and CD16+ monocyte) were still enriched in EPCR+ cells while CD90+EPCR−/MPP cells were enriched in the CD10+ B cell-associated gene module (Fig. 6b), with a considerable number of the 547 and 2154 DE genes found within these (and others; not shown) lineage-primed modules (Fig. 6c). Indeed, 5.6 and 4.6% of the 547 and 22.7 and 16.4% of the 2154 DE genes were located in the Velten's[8] and Oetjen's[9] gene modules, respectively (except those related to HSC/Progenitors/HSPC gene sets; see Supplementary Data 4). Interestingly, the normalized

expression of these DE genes associated with these modules was high in these primitive populations (red to light blue representation in the heatmaps; Fig. 6c). This suggests that human HSPCs harbour considerable levels of lineage-priming programs and the difference in the quality of lineage output between these populations may be governed by the concomitant reduction in the expression levels of the multilineage/myeloid gene sets with the increase in lymphoid module(s) when EPCR+ cells differentiate into CD90+EPCR−/MPP cells.

In summary, all the HSC features were found to be significantly enriched in EPCR+ cells compared to CD90+EPCR− progenitors, and MPPs, thus supporting our hypothesis that EPCR+ cell population is highly enriched in bona-fide HSCs, and the multipotent features of such cells were confirmed functionally in vivo.

**EPCR+ cells represent a highly homogeneous HSC population.** Following these findings, we went on to determine if additional heterogeneity may exist within EPCR+ HSCs by analyzing the expression of cell surface antigens described to be associated with human HSPCs such as CD33, CD133, CD143, and CD93. We found that these antigens were highly expressed on most EPCR+ HSCs but not in downstream progenitors (Supplementary Fig. 4). We then generated single cell expression data using the C1 Fluidigm system in order to characterize residual variability

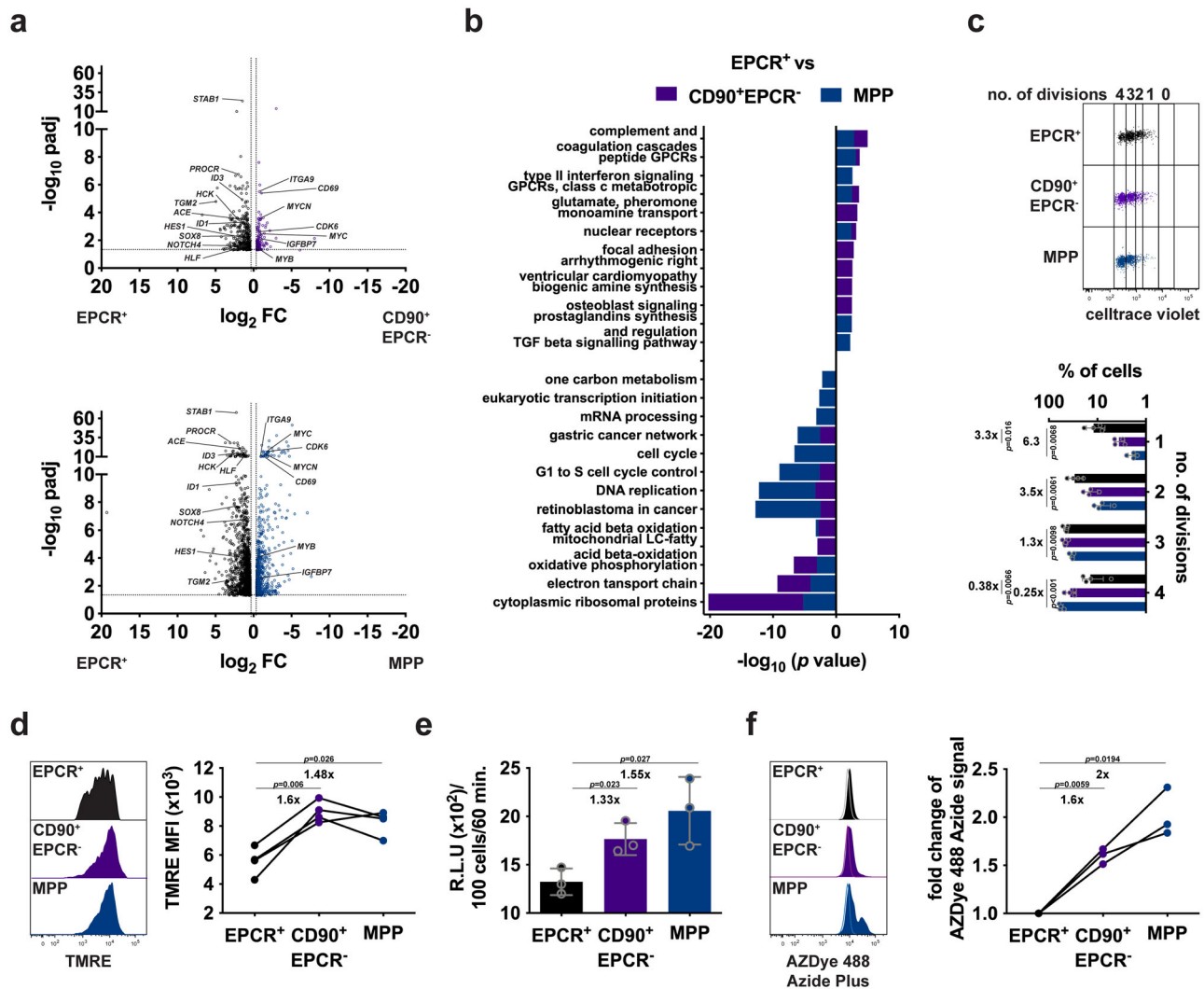

**Fig. 5 Defining the molecular features of EPCR⁺ HSCs. a** Volcano plot representation of the RNA-seq data derived from 5 EPCR⁺ vs 4 CD90⁺EPCR⁻ (top) or vs 5 MPP samples (bottom). **b** Wikipathways analysis identified significant enriched functional categories (FDR < 0.05 with P-values adjusted using Benjamini–Hochberg method). **c** Cell division tracking of the denoted populations (n = 4) at day 4 cultured in serum-free condition with a representative FCM analysis of the cells. **d** Mitochondria membrane potential of the denoted HSPCs determined by TMRE staining (n = 4) with a representative FCM analysis. **e** ATP production by the denoted cells determined with the CellTiter-Glo® luminescent assay (n = 3). **f** Protein synthesis detected in the indicated HSPCs assayed with an OP-Puro protein synthesis kit (n = 3) with a representative FCM analysis. Open and filled histograms represent the signal obtained without (w/o) and with (w) OP-Puro treatment respectively. Normalized values (to the signal derived from EPCR⁺ HSC) are shown. The CD201 clone RCR-401 antibody was used in all the experiments. Bars are the mean values and error bars are the S.D for the no. of experiments performed. Student t-test (two-tailed unpaired in **c**, **e** and two-tailed paired in **d**, **f**) was used. Source data are provided as a Source Data file.

within EPCR⁺ HSCs. Sparse factor analysis[20] in 135 QC-passed single EPCR⁺ HSCs (with 3 million reads/cell) identified 18 genes as the main sources of residual cell–cell variability (Supplementary Fig. 5). We visualized cellular variability via a t-SNE-based embedding and PCA, and this did not reveal evident grouping structure (Fig. 7a and Supplementary Fig. 5). In addition, hierarchical clustering also showed great similarity in the overall gene expression profile between these single cells (> 90% of the cells; Fig. 7b). Taken together this suggests little transcriptional heterogeneity/substructure between EPCR⁺ HSCs.

To further probe for cell–cell heterogeneity, we performed single cell functional assays. First, we verified the in vitro multilineage capacity of EPCR⁺ HSCs at the population level (Supplementary Fig. 6a). In addition, we uncovered that all the early hematopoietic lineage-priming modules[8] analyzed (even the megakaryocyte, erythroid/megakaryocyte and erythroid modules; not shown) were expressed in EPCR⁺ HSCs (Supplementary

Fig. 6b), thus supporting again their multilineage properties. We then uncovered that single EPCR⁺ HSCs gave rise to 56% of clones with CD34⁺ HSPC output, thus suggesting their primitive nature, and more importantly 80% of the clones were multipotent (Fig. 7c, e). This again was the highest frequency of multipotent output by an HSC enriched-population that has been previously described in the literature[6,7,10]. Indeed, others that investigated and/or attempted to use a sub-fraction of the CD90⁺CD49f⁺ population were able to show that at best ~40–50% of their testing single cells were multipotent[6,7,10]. In addition, when investigating the differentiation priming at the single cell level, a high homogeneity of gene expression profile between the single cells were also detected (Fig. 7f). Notably, when we determined the gene expression strength (the overall weight expression of the genes comprised in each gene module) the same multilineage- and neutrophil-associated modules had the strongest expression in single cells as compared to the whole EPCR⁺ population

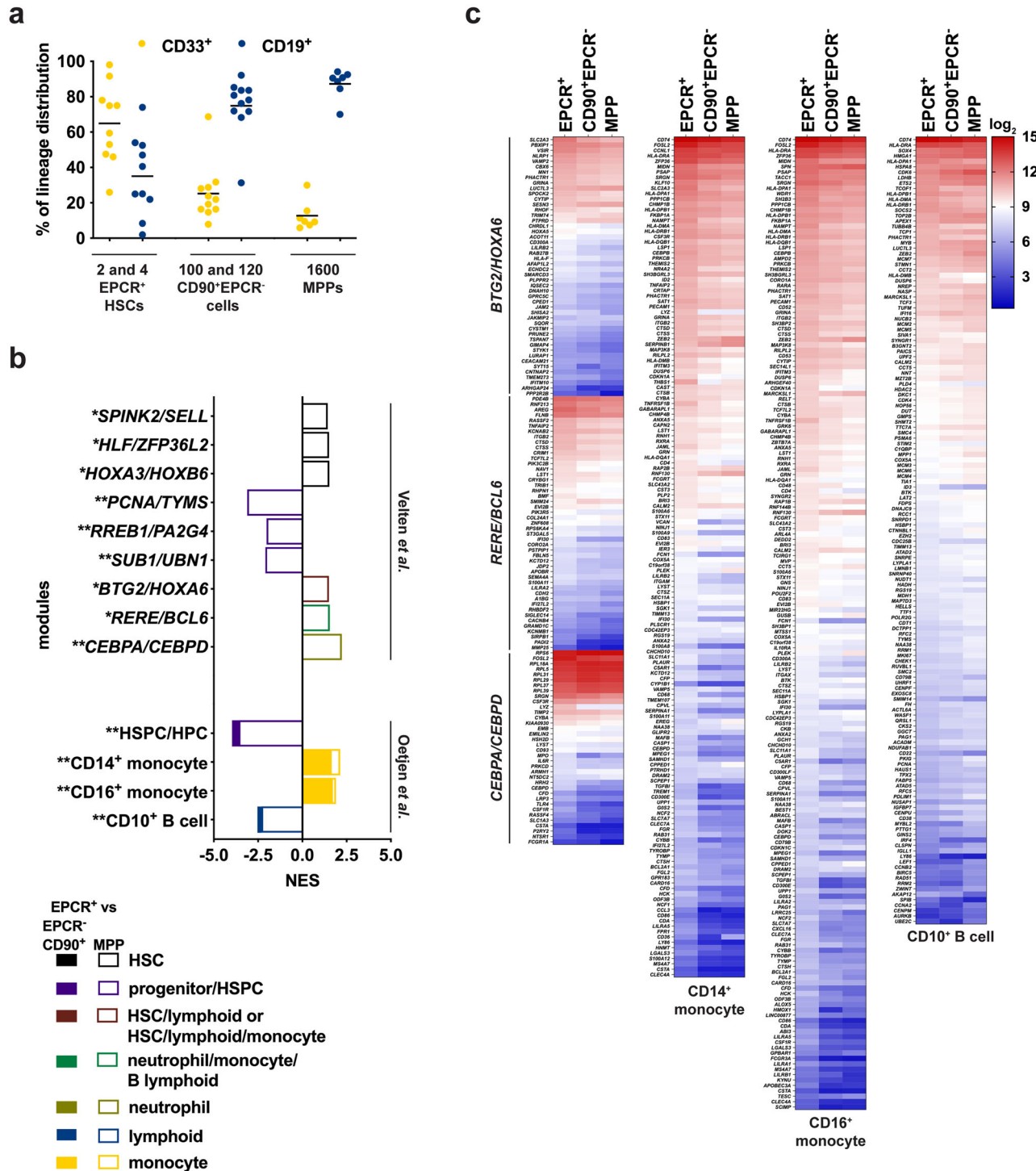

**Fig. 6 Unique transcriptomic priming supports a in vivo balanced multilineage output of EPCR+ HSCs. a** In vivo human myeloid (CD33+) vs B lymphoid (CD19+) differentiation outputs of EPCR+, CD90+EPCR- and MPP cells in 1ry NSG mice at 12 wks post transplanted with the indicated cell doses ($n = 7–11$ mice); mean lines are shown. **b** Denoted lineage-priming gene modules[8,9] enriched in EPCR+ HSCs, CD90+EPCR- progenitors, and MPPs; NES, normalized enrichment scores enriched (FDR p-value *<0.1 and **< 0.05 by pre-ranked Gene Set Enrichment Analysis, GSEA; p-values were adjusted using Benjamini–Hochberg method) are shown. **c** Heatmap of DE genes (padj <0.05 and with cut-off of log2FC ± 0.4, see Supplementary Data 4) between EPCR+ vs EPCR-CD90+ vs MPP found in the enriched lineage-priming gene modules shown in (**b**). The scale represents log2 of normalized transcript expression values. Source data are provided as a Source Data file.

(Fig. 7g and Supplementary Fig. 6b). To date, these findings strongly suggest that EPCR+ cells represent a human HSC population that is inherently harbouring a high multipotential ability and homogenous transcriptomic profile.

## Discussion

The use of CD49f expression has undeniably helped to highly enrich human HSCs from cord blood. However, the CD90+CD49f+ population is still heterogeneous, and due to the weak CD49f

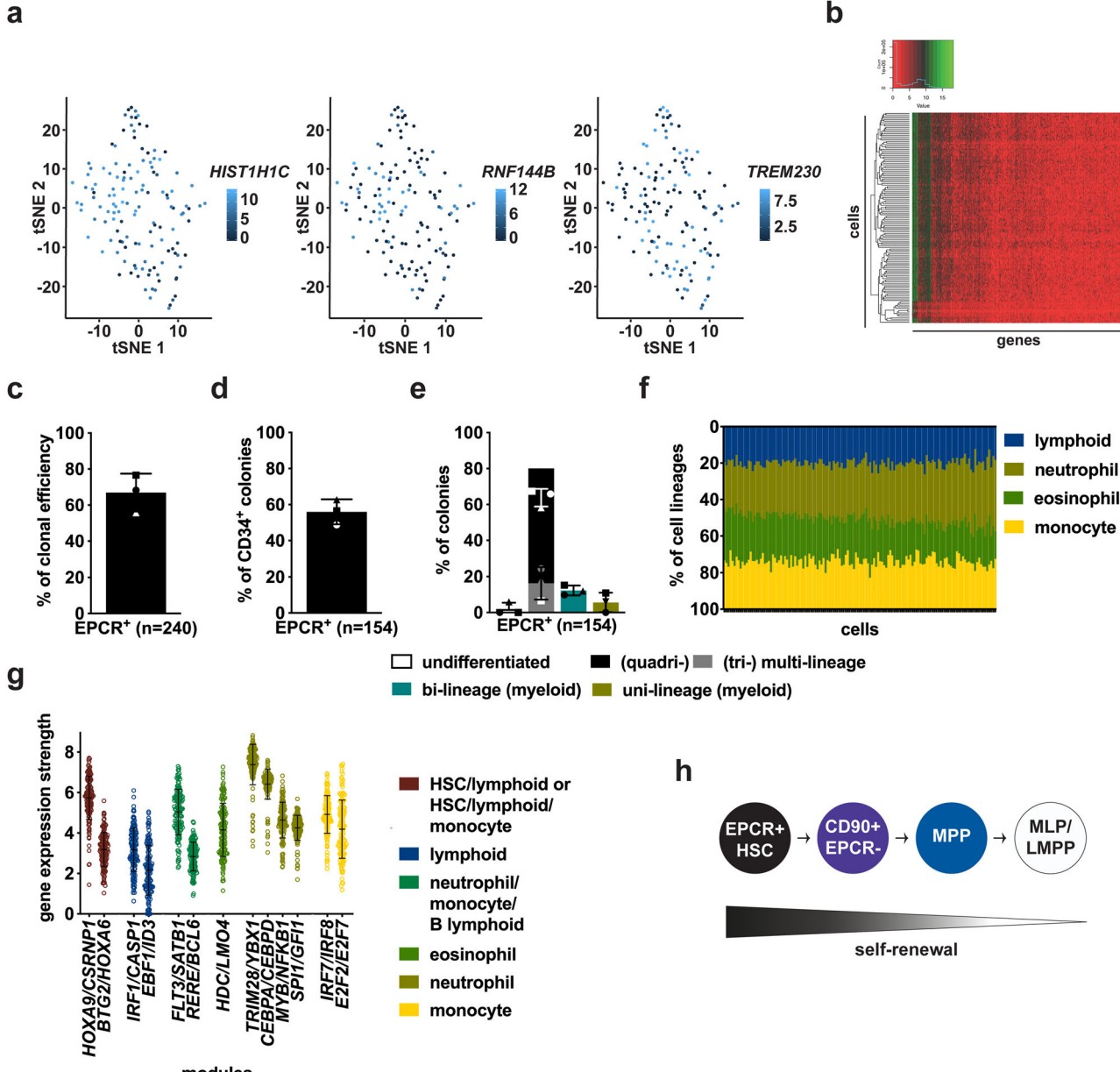

**Fig. 7 Single cell transcriptomic and functional studies show that EPCR$^+$ is a quasi-homogenous HSC population. a** Representative t-SNE-based plots of 135 single EPCR$^+$ HSCs based on the most variable genes determined in Supplementary Fig. 6. **b** Hierarchical clustering of 135 single EPCR$^+$ HSCs. **c** Clonal efficiency of 240 single EPCR$^+$ cell cultures (from three independent experiments). **d** Percentage of colonies containing CD34$^+$ HSPCs derived from single EPCR$^+$ HSCs ($n = 154$ colonies from three independent experiments). **e** Percentage of colonies of the indicated type derived from single EPCR$^+$ HSCs ($n = 154$). **f** Distribution of the overall gene expression in 135 single EPCR$^+$ HSCs associated with the denoted cell lineages based on the lineage-priming modules used[8]. **g** Expression strength of the genes associated with the denoted lineage-priming modules detected in the single EPCR$^+$ HSCs ($n = 135$). **h** Schematic representation of the new hierarchical organization within the most primitive CD34$^+$ compartment supported by all the data presented in this study. Bars and lines are the mean values and error bars are the S.D for the no. of experiments or single cells performed. Source data are provided as a Source Data file.

expression on human CD34$^+$ HSPCs, this population contains variable numbers of CD49f$^-$ cells depending on how stringent each investigator defines the positivity of CD49f expression. This has impacted on data reproducibility between different laboratories and on data interpretation but also when assessing the transcriptional profile at the single cell level. Indeed, it was recently reported that human HSPCs appear to have a "CLOUD" structure[8] where the authors showed the existence of a low-priming status of the human HSC-enriched and MPP populations that resulted in the lack of an hierarchical tree-structure within these primitive cell populations at

the transcriptional level despite their differential functional properties. This biological interpretation could have resulted, at least in part, from a very poor resolution of the human CD90$^+$CD49f$^+$ population from adult BM. In an attempt to further enrich human HSCs from the CD90$^+$CD49f$^+$ population, Eaves' group showed that within this barcoded population, cells that highly express a commonly known myeloid antigen CD33 harbour the highest self-renewal capacity[7]. Despite their efforts, the authors of this work did not further show HSC enrichment of their testing population.

To solve this, we went on dissecting the potential relationship between the different primitive CD34[+] HSPCs and uncovered that: (1) the CD90[+]CD49f[+] HSC-enriched fraction does not encompass all human HSCs as we have identified another HSC-enriched population (CD90[−]CD49f[+]) and both populations are interconvertible and give rise to the whole CD34[+] HSPCs; (2) both CD90[+]CD49f[+] and CD90[−]CD49f[+] HSC-enriched populations are still heterogenous and scRNA-sequencing and functional assays demonstrate that only a fraction of cells (~6–10%) within these two populations have long-term repopulating capacity; (3) using transcriptomic and extensive in vivo studies we have identified a primitive human hematopoietic population, CD34[+]EPCR[+](CD38/CD45RA[−]), and this population contains all the cells with robust repopulating capacity within the two CD49f[+] HSC-enriched populations; (4) EPCR[+] HSCs are multipotent with a unique multilineage-primed transcriptomic wiring; (5) EPCR[+] HSCs harbour the highest human HSC frequency identified to date, ~1 in 3 cells, which represents a significant improvement from 1 in 10–20 CD90[+or−]CD49f[+] cells, thus closely matching the purities described in mice;[18,19] (6) EPCR[+] HSCs are transcriptomically and functionally very homogeneous at the single cell level and can be easily identified in adult BM; (7) human primitive HSPCs have a well-structured cell lineage relationship where the lymphoid-bias CD90[+]EPCR[−] progenitor cell population, downstream of EPCR[+] HSCs, gives rise to MPPs.

Although EPCR has been found to be expressed on human cord-blood expanded cells with UM171[13,14], all these works have only demonstrated an association of the percentage of EPCR[+] cells in the heterogeneous expanded HSPC population with the capacity of having more repopulating cells, and the fact that UM171 was able to maintain the presence of EPCR expression in their culture conditions. Moreover, none of these studies have actually formally demonstrated the existence of a highly purified human HSC population-based on EPCR or another cell surface marker on unmanipulated cells, and most importantly that this same population also exists in adult BM. Therefore, our data demonstrate that not only EPCR[+] cells represent a bona-fide primitive HSC population with multipotent and self-renewal capacity but also our study is able to provide the highest resolution of a human HSC population described to date. Importantly, we demonstrate that the lymphoid-bias CD90[+]EPCR[-]cell population precedes MPP/MLP cells. As such, we were able to resolve and refine the human HSPC hierarchical structure (Fig. 7h) both transcriptionally and functionally. In fact, another potential important observation from our study is that human EPCR[+] HSCs and its immediate progenies appear to have considerable levels of lineage priming. This seems to be contrary to what has been described that human HSCs are low-primed transcriptomically[8]. This could be a significant finding and warrants future investigations. It would also be important to further investigate how the different gene modules that are associated with lineage commitment/differentiation are regulated (e.g., epigenetically) in the near future, as these studies may lead to a better understanding of human HSPC regulation.

Overall, we strongly believe that the ability to isolate and further characterize these highly enriched HSCs opens the avenues to further understand how the human hematopoiesis is regulated during development, homoeostasis, and regeneration, as well as in age-related processes such as clonal haematopoiesis and leukemogenesis.

## Methods

**Cell processing**. Cord blood was obtained after informed consent from the Royal London Hospital (REC: 06/Q0604/110) and University of Hospitals of Wales (REC: 06/WSE03/6). The protocols were approved by their respective ethical committee. Human bone marrow MNCs were purchased from Lonza Biologics and StemCell Technologies and CD34[+] cells were obtained as described for CB mononuclear cells (MNCs). Informed consent was obtained in accordance with the Declaration of Helsinki. Unless stated otherwise, three or more samples were pooled for each experiment and MNCs were obtained by density centrifugation using Ficoll-Paque. CD34[+] cells were positively selected using EasySep CD34[+] selection kit. Lineage depleted (Lin[−]) cells were obtained using StemSep Human Progenitor enrichment cocktail kit according to the manufacturer's protocol.

**Immunophenotyping and cell sorting**. In general, a maximum of $1 \times 10^6$ cells were incubated with antibodies (4 µl each) for 30 min at 4 °C in 50 µl of staining buffer (phosphate buffer saline, PBS with 2% of foetal bovine serum, FBS), washed and resuspended in DAPI (1:2000 from a 200 µg/ml stock) containing solution before analysis on an LSRFortessa (BD Biosciences) flow-cytometry (FCM) or cell sorting. Gates were set up to exclude non-viable cells, debris, and doublets. The FlowJo software was used for FCM analysis. Cell sorting was performed at a flow rate of 2000–4000 events/sec on purity mode. Viable cells were double-sorted and purity checked afterwards. Cell counts were performed in an automated cell counter to attain accurate cell counts for the LDA assays. To stain cells derived from MS5 co-cultures, these were first detached/collected then the content of each well was stained in 20 µl of staining buffer containing 1 µl of each antibody.

**Cell division tracking, mitochondria stains, ATP, and protein synthesis assays**. CellTrace[TM] Violet staining kit was used to label CD34[+] HSPCs according to the manufacturer's instructions. Briefly, $1 \times 10^6$ cells were incubated with the dye at 5 µM in PBS at 37 °C for 20 min. Cells were washed with PBS with 2% FBS followed by cell surface immunophenotyping as described for cell sorting. Then, 500–1000 of the sorted cells were cultured in 100 µl of StemSpan[TM] SFEM II media containing of SCF (300 ng/ml), FLT3L (300 ng/ml), and TPO (20 ng/ml) for 4 days before FCM analysis. To determine mitochondria membrane potential TMRE (tetramethylrhodamine ethyl ester) stain was performed. Briefly, $1 \times 10^6$ cells/ml were incubated with the dye at 20 nM in PBS at 37 °C for 20 min. Cells were washed with PBS with 2% FBS followed by cell surface immunophenotyping. CellTitre-Glow 2.0 kit was used to assay the basal ATP production by the different HSPCs. One hundred FACS sorted cells were incubated in StemSpan[TM] SFEM II media with 2% FBS at 37 °C for 60 min followed by cell lysis and luminescent detection according to the manufacturer's instructions. To determine protein synthesis Click-&-Go Plus 488 OPP Protein Synthesis Assay Kit was used. Briefly, CD34[+] cells ($1 \times 10^5$ cells in 100 µl of StemSpan[TM] SFEM II media containing of SCF (300 ng/ml), FLT3L (300 ng/ml), and TPO (20 ng/ml)) were pre-stimulated at 37 °C for 2 h followed by an hour of incubation with either vehicle or 30 µM of OPP reagent. After this, cells were subjected to cell surface immunophenotyping, fixed with 2% of methanol-free formaldehyde (in PBS) at RT for 15 min, permeabilized with 0.1% saponin/3% FBS solution (in PBS) at RT for 10 min, and stained with Azide-AlexaFluor488 according to the manufacturer's instructions.

**Xenotransplantation**. NSG (NOD.Cg-Prkdcscid Il2rgtm1Wjl/SzJ) mice were originally obtained from The Jackson Laboratory and bred in isolators with aseptic standard operation procedures in the Biological Research Facility of The Francis Crick Institute. Once weaned, mice were kept in individual ventilated cages. Mice were kept at 18–23 °C with 40–60% humidity and under a 14/10 h of light/dark cycle. All animal experiments were performed under the U.K Home Office project licence (70/8904) in accordance with The Francis Crick Institute animal ethics committee guidance. NSG mice aged 8–12 weeks were irradiated at 3.75 Gys ([137]Cs source) up to 24 h before tail vein injection. In addition, at least 8 h before transplantation mice were intraperitoneally administered with 0.5 mg/g of IVIGs. We have previously shown that this procedure reduces phagocytosis of antibody labelled human cells by the residual murine innate immune cells[21]. Purified cell populations were mixed with 250,000 irradiated (15 Gys) CD34[−] MNCs as accessory cells prior to injection into each mouse (in PBS with 2% of FBS and penicillin/streptomycin). Mice were supplied with acidified drinking water one week before myeloablation and throughout the whole experiment. Mice were also administrated with antibiotics (e.g., intraperitoneal injection of Betamox) every other day for the first two weeks after adoptive transfer. These small adjustments such as irradiation dose and administration of antibiotics and IVIg were critical that ensured enhanced human engraftment with limited cell dose and survival of the mice. In most experiments, especially in the LDA assays, similar number of male and female mice were used to avoid potential bias. For secondary transplants, groups of mice were sacrificed by cervical dislocation and bone marrow (BM) was harvested from six bones (tibiae, femurs, and ileum) and pooled for depletion of mouse cells using EasySep Mouse/Human Chimera Enrichment kit according to the manufacturer's instructions. Recovered human cells were immunophenotyped and sorted for human CD45[+]CD34[+]CD38[−] HSPCs and transplanted alongside with 300,000 of FCM purified and irradiated (15 Gys) human CD45[+]CD34[−] cells/mouse.

**Analysis of mouse bone marrow**. Animals were sacrificed at different time points indicated and the cells from the BM were harvested. Red blood cells were lysed and an aliquot of BM cells were immunophenotyped to determine the level and nature of human engraftment by FCM. Dead cells and debris were excluded using DAPI staining and >200,000 DAPI- events were collected. Human lympho-myeloid engraftment was quantified as the proportion of live cells that were CD45[+]CD19[+] and CD45[+]CD33[+]. It was considered positive for human engraftment when

>0.01% of human cells with clear myeloid and lymphoid sub-populations were detected. SRC frequency was calculated using the extreme limiting dilution analysis software (ELDA). Due to their rarity, quantification of the different human HSPCs engraftment was achieved by staining all the cells collected from the six bones of a mouse and by acquiring the maximum number of human CD45+ events possible and marrows with 200–300 or more engrafted CD34+CD38− HSPCs were considered for analysis.

**MS5 long term cultures (LTC).** MS5 (DSMZ) cell line was cultured in αMEM supplemented with 10% of FBS and 1% of P/S. Cells were split 1:4–1:6 every 3 days and cells between passage 3–6 after thawing were used for long-term cultures (LTCs). Before cell seeding, 96-well plates were coated with collagen diluted 1:10 in PBS (0.3 mg/ml). To coat the wells, 35 μl of the diluted collagen solution was added to each well and was left for 1 min. The plates were left to dry for ~1 h and then washed two times with PBS to neutralize the collagen pH. Then, 3000 MS5 cells/well were plated and cultured for 2 days until 80–90% confluence was reached. A day before cell seeding, plates were irradiated at 7.5 Gy to arrest cell growth. Culture medium was changed 3–4 h after irradiation to 200 μl/well of MyeloCult™ H5100 with 1% P/S containing the following human cytokines: SCF (40 ng/ml), FLT3L (20 ng/ml) and IL7 (20 ng/ml). Half media change containing the same cytokines was performed every week and the last media change G-CSF (20 ng/ml), IL2 (10 ng/ml), and IL15 (10 ng/ml) were also included. LTCs were maintained for 4 weeks before analysis. This optimized cytokine cocktail condition not only allows the proliferation of hematopoietic cells and adequate B lymphoid outputs but also avoids an overproduction of myeloid and NK-like cells that competes with B lymphoid differentiation if all the cytokines were present throughout the 4-week culture[22]. For single cell cultures, EPCR+ HSCs were sorted directly onto MS5 cultures and were cultured as described.

**Total RNA extraction, library preparation, and RNA-sequencing.** Total RNA was extracted using RNeasy Plus Micro Kit and quality was verified in an Agilent 2100 Bioanalyzer (Agilent) and samples with RNA integrity number (RIN) of 8 or above were used prior to library preparation. Libraries were prepared using KAPA Stranded with RiboErase RNA-seq kit (according to the manufacturer's instructions). Briefly, 20–25 ng of starting RNA were subjected first to cytoplasmic and mitochondrial ribosomal RNA (rRNA) depletion by hybridization of complementary DNA oligonucleotides, followed by treatment with RNase H and DNase to remove rRNA duplexed to DNA and original DNA oligonucleotides. Samples depleted of rRNA were then subjected to 94 °C for 6 min in the 2× Fragment, Prime, and Elute Buffer in order to obtain 200–300 bp fragments. cDNA synthesis was run in two steps following the manufacturer's instructions. The ligation step consisted of a final volume of 110 μl of the adaptor ligation reaction mixture with 60 μl of input cDNA, 5 μl of diluted adaptor, and 45 μl of ligation mix (50 μl of ligation buffer + 10 μl of DNA ligase). The Kapa Dual-Indexed Adaptors stock concentration was diluted to 1.5 mM to get the best adaptor concentration for library construction. The ligation cycle was run according to the manufacturer's instructions. To remove short fragments such as adaptor dimers, 2X AMPure XP bead clean-ups were done (0.63 SPRI and 0.7 SPRI). To amplify the library, 9 PCR cycles were applied to the cDNA KAPA mix. Amplified libraries were purified using AMPure XP. The quality and fragment size distributions of the purified libraries were assessed by a 2200 TapeStation Instrument (Agilent Technologies, USA). Sequencing was then performed in a HiSeq2500 or HiSeq4000 Sequencing System (Illumina) with 30 million single-end 100 bp reads/sample.

**Single cell RNA extraction, library preparation and RNA-sequencing.** The experiments concerning the two CD90+or−CD49f+ populations, single cells from 3 single cord blood were sorted directly into 96-well skirted FrameStar PCR plates with 2.3 μl/well of 0.2% solution of TrixonX-100 supplemented with 1 U/μl of RNAse inhibitor. NGS library was constructed with Nextera XT DNA Sample Prep kit. Multiplexed Nextera XT libraries were prepared according to the manufacturer's protocol (Illumina).

The Fluidigm microfluidics system was used to isolate single CD34+CD38−CD45RA−CD49f+EPCR+ cells in double-filtered RNase-free PBS containing 0.5% of FBS (300–500 cells/μl). Sorted cells were mixed with C1 Cell Suspension Reagent before loading onto a 10–17 μm-diameter C1 Integrated Fluidic Circuit (IFC; Fluidigm). LIVE/DEAD staining solution was prepared by adding 2.5 μl of Ethidium homodimer-1 and 0.625 μl of Calcein AM (Life Technologies) to 1.25 ml of C1 Cell Wash Buffer (Fluidigm) and 20 μl of this mix was loaded onto the C1 IFC. Each capture site was carefully examined under a Nikon microscope in bright field, GFP, and Texas Red channels for cell doublets and viability. Cell lysis, reverse transcription, and cDNA amplification were performed on the C1 Single-Cell Auto Prep IFC, as specified by the manufacturer (protocol 100-7168 E1). The SMART-Seq v4 Ultra Low Input RNA Kit for the Fluidigm C1 System was used for cDNA synthesis from the single cells. Illumina NGS library was constructed with Nextera XT DNA Sample Prep kit. Multiplexed Nextera XT libraries were prepared manually following an altered manufacturer's protocol. Briefly, samples were normalized to 0.2 ng/μl (1 ng total) of DNA material per library. Each 5 μl of the tagmentation reaction mixture consisted of

2.5 μl of TD buffer, 1.25 μl of input DNA, and 1.25 μl of ATM (amplification tagment mix). Then, 1.25 μl of Stop solution NT was used after Tagmentation step. For the amplification of the library, 3.75 μl of NMP reaction mix was added, plus 1.25 μl of each Nextera XT Index Kit (96 indexes; Illumina). The number of cycles applied for the library construction were 12 according to the manufacturer recommendation for no less than 1 ng of input DNA. Amplification was performed by Veriti 96-well PCR (Applied Biosystems) followed by 2 rounds of 0.9× AMPure XP bead cleanup. The quality of the purified libraries was assessed using an Agilent High Sensitivity DNA Kit (Agilent). Pooled libraries were denatured and diluted to 5.5 pM and clonally clustered onto a pair-end patterned flowcell using the HiSeq 4000 PE Cluster Kit on the cBot (Illumina). Sequencing was set for 3 million 100 bp pair-end reads per cell under Illumina's kit instructions.

## Bioinformatic analysis

*Bulk population RNA-sequencing.* Sequenced read fragments were mapped to the human reference genome GRCh38 using STAR (STAR_2.4.0 h)[23]. Expression counts estimates were generated using RSEM[24]. DESeq2[25] was used to test for differential expression and the results were considered significant at an FDR of 0.1. PCA visualization was performed based on variance stabilized counts using the DESeq2 package. For EPCR+ and CD90+EPCR− populations, we performed pathway analysis using a competitive gene set test accounting for inter-gene correlation[15] and computed normalized enrichment scores using the fgsea R package[26]. Pathway enrichment (WikiPathways) was considered significant at an FDR of 0.05.

To study lineage-priming we performed gene enrichment (using GSEA platform), gene expression strength and other analyses using the DE gene lists between the 3 HSPC populations (Supplementary Data 4) and applied onto the gene set modules determined/defined by Velten et al.[8] (for early cell lineage differentiation) and Oetjen et al.[9] (differentiated cells) as our gene set database. For GSEA, we used pre-ranked DE gene lists (*p*adj < 0.1). The gene expression strength is the overall weight expression of the genes comprised in each gene set or module. This was done by regressing the normalized counts of the genes in each module but taking in consideration the overall gene expression found in each population.

**Single-cell RNA-sequencing.** Raw sequencing reads were mapped with STAR, using the human reference genome (GRCh38). Expression counts estimates and transcripts per million (TPM) were generated using RSEM. Following the counting of mapped reads, we imposed additional quality control criteria to identify and remove low quality cells. Cells that did not pass all of the following criteria were removed from the analysis: 1-total number of reads >50,000; 2-number of genes detected (at least one mapped read) >2000; 3-percentage of mitochondrial reads <15%.

For CD90+or−CD49f+ cells, 469 out of 516 single cells passed all criteria. To assess the effect of technical noise, we first used a log-linear fit to capture the relationship between mean and squared coefficient of variation (CV) of the log-transformed, TPM data[27]. We then considered genes with a squared CV greater than the estimated squared baseline CV as variable beyond technical noise. This filter for highly variable genes resulted in 3754 genes, which were augmented by the candidate genes identified from the total RNA-seq analysis and used for downstream analysis, resulting in a total of 6503 genes. To account for possible confounders, including batch effects, we applied SVA[28] with one component to the TPM data and regressed out the resulting confounding factor. To further remove any remaining batch effects whilst controlling for experimental design (cell type), we applied combat[29]. As quality control that batch effects were accounted for, we used a 2D PCA of the residual expression values. After correcting for confounders, we found good mixing between all batches.

Downstream analyses were conducted on an informed gene set, retaining only the candidate genes that were identified in the total RNA-seq analysis. To visualize single cells and to explore the presence of possible sub-populations, we first generated a 2D heatmap using all 469 cells (using the R package gplots), and we applied PCA as well as a t-SNE to visualize the cells using 2D coordinates (using R package Rtsne)[30]. With the lack of any distinct sub-population suggesting a more continuous difference between the two sub-populations, we computed a diffusion map representation to capture this potential continuum. The diffusion map then revealed a transition from cells enriched in CD90+CD49f+ cells (SP1) to cells enriched in CD90−CD49f+ cells (SP3), with a common sub-population in between (SP2). To characterize this common sub-population, we performed a differential expression analysis, contrasting SP1 and SP2 and SP3 and SP2 respectively, using the Wilcoxon rank test. This revealed a set of 950 genes that were differentially expression (with *p*adj < 0.1 and with cut-off of log$_2$FC ± 0.8) in both comparisons.

For CD34+CD38−CD45RA−CD49f+EPCR+ cells, 135 out of 183 cells passed all QC criteria. As before, we used a log-linear fit to capture the relationship between mean and squared CV of the log-transformed, TPM data. 3122 highly variable genes were used for downstream analysis. To explore whether EPCR+ cells could be further sub-divided, we again performed a heatmap and t-SNE analysis, but could not find evidence for the existence of any sub-population. To disentangle the source of variation driving the observed cell–cell variability we fitted a salmon model[20] using the 50 MSigDB core processes[31], augmented with the lineage-priming pathways modules[8].

**Statistical analyses**. The GraphPad Prism was used for all statistical analyses except for RNA-seq. Unless otherwise indicated in figure legend, mean ± SD values are reported in the graphs. Statistical significance was determined using Student (two-tailed) paired or unpaired t-tests. To generate DE genes between the different populations analyzed, negative binomial generalized linear model (GLM) fitting and Wald statistics, computed using the DESeq2 R package with standard settings, were used. The $\log_2$FC was divided by lfcSE, which was compared to a standard normal distribution to generate a two-tailed p-value. P-values were then adjusted using Benjamini–Hochberg method. For Wikipathways analyses correlation adjusted MEan RAnk (CAMERA) gene set test with standard settings as implemented as part of the limma R package[15] was used. It tested whether the genes in the set were highly ranked in terms of DE relative to genes not in the set and accounted for inter-gene correlation. P-values were then adjusted using Benjamini–Hochberg method.

Normalized enrichment scores (NES) were calculated using the online GSEA software. The scores were normalized to mean enrichment of random samples of the same size. An adaptive multilevel splitting Monte Carlo approach was used to compute p-values as implemented in the fgsea R package. P-values were then adjusted using Benjamini–Hochberg method.

**Reporting summary**. Further information on research design is available in the Nature Research Reporting Summary linked to this article.

## Data availability

The accession numbers for the RNA-seq data in this manuscript are: GSE154588 (RNA-seq of the four CD90$^{+or-}$CD49f$^{+or-}$ cells) and GSE154931 (scRNA-seq of the two CD49f$^+$ cells), GSE155174 (scRNA-seq of EPCR$^+$ HSCs), GSE154263 and GSE197079 (RNA-seq of EPCR$^+$, CD90$^+$EPCR$^-$ and MPP cells). All the RNA-seq analyses generated in this study are provided in the Supplementary Data and in the Source Data file. Wikipathways database was used (https://www.wikipathways.org/index.php/Download_Pathways) in this study. Source data are provided with this paper.

## Materials availability

Figures were assemble using Adobe Illustrator and detailed information on other materials and reagents is listed in the Supplementary Information.

## Code availability

The DeSeq2 code is available at https://bioconductor.org/packages/release/bioc/html/DESeq2.html, STAR at https://github.com/alexdobin/STAR, RSEM at https://github.com/deweylab/RSEM and R at https://www.r-project.org. Online software such as ELDA (http://bioinf.wehi.edu.au/software/elda/index.html) and GSEA (www.gsea-msigdb.org/gsea/index.jsp) were used in this study.

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

## Acknowledgements

F.A.-A. would like to specially thank The Jane Hodge Foundation for their support. We thank the Biological Resource Unit (BRU), the Flow Cytometry, the Advanced Sequencing core facilities at the Francis Crick Institute. This work was supported by The Jane Hodge Foundation and the UK Biotechnology and Biological Sciences Research Council (BB/S017097/1; to F.A.-A.) and the Francis Crick Institute (to D.B.), which receives its core funding from Cancer Research UK (FC0010045), the UK Medical Research Council (FC0010045), and the Wellcome Trust (FC001045). For the purpose of Open Access, the author has applied a CC BY public copyright licence to any Author Accepted Manuscript version arising from this submission.

## Author contributions

F.A.-A. conceptualized, designed, and interpreted all experiments and data, executed all experiments except cell sorting and RNA-seq library preparation and sequencing, and wrote the paper; F.B. performed all the transcriptomics analyses; S.M. performed experiments and helped with data interpretation; H.R. performed cell sorting; J.P.-L. performed RNA library preparation and sequencing; M.G.-A. and N.R. helped with data interpretation; L.A.-M. helped with mouse work and mouse colony management; D.B. designed and interpreted experiments. All authors helped with the editing and approved the manuscript. F.A.-A. and D.B. contributed as co-senior authors.

## Funding

## Competing interests

The authors declare no competing interests.
