## [Peer Review File · Nature Communications]

Single Cell Analyses Identify a Highly Regenerative and Homogenous Human CD34+ Hematopoietic Stem Cell PopulationREVIEWER COMMENTS

Reviewer #1 (Remarks to the Author):

The manuscript submitted by Bonnet et al. demonstrated that a specific subpopulation of human HSPCs (CD34+) labelled by EPCR have high repopulating and self-renewal abilities, as well as being transcriptomically and functionally homogenous via single cell analysis. Although the study can provide reasonable evidence, the primary concern is that features of EPCR in human HSCs have been well characterized in past few years (Fares et al, 2017; Subramaniam et al, 2019), hence the data presented serve as good supporting evidence in another viewpoint, but lacks originality to meet the novelty criteria.

More specific comments are listed below.

Major concerns

1. Here the study defines EPCR+ (vs CD90+EPCR-) populations of human HSCs and proposed a hierarchy of self-renewal capability, this is a nice viewpoint, but doesn't seem to extend the HSC characterization methods significantly, with regard to existing studies and evidence available.

2. Single cell analysis demonstrating that EPCR+ HSCs is homogenous population is reasonable, but it would be much more interesting and informative if single cell analysis was also performed on more heterogenous population such as CD34+CD38-CD45RA- HSPCs. Even the scRNA-seq result in Figure 2b isn't informative enough to confer distinct signatures among the 2 populations. If some outstanding clusters that specifically express EPCR can be identified, further analysis may reveal more unique signatures associated with EPCR+ clusters.

3. Most assays were performed on EPCR+ vs CD90+EPCR- populations, it would be more comprehensive if CD90-EPCR- population identified as MPPs were also included in the analysis (Figure 4,5, 6).

Minor concerns

1. How well is CD49f expression correlating to EPCR expression in CD34+ cells? Authors may consider to plot CD49f vs EPCR to examine the patterns.

2. Have the authors tried t-SNE or UMAP representation to visualize in Figure 2b?

3. Figure 2d that is described in the manuscript went missing in the actual figure.

4. Figure 5, it will be more informative if the author could perform a few additional functional assays to validate EPCR+ vs CD90+EPCR- HSC features based on GO analysis. Functional assay that may be important for validation includes OP-puro protein synthesis (ribosomal proteins), seahorse XF or cell tilter glo (OXPHOS).

Reviewer #2 (Remarks to the Author):

Comments to the authors:

Anjos-Afonso et al. present in this manuscript a new method to isolate with high purity human hematopoietic stem cells (HSCs). This is a very important contribution to the field. The transplantation of HSCs is a life saving procedure for many patients across the world. Considering that sources of HSCs are limited and therefore very precious, being able to isolate HSCs with the highest purity would ensure that they are best used in medical applications. Here the authors used flow cytometry to characterize and isolate HSCs and combine it to single-cell transcriptomics to identify new cell surface markers characteristic of human HSCs. This is how they found the marker EPCR which allowed them to isolate with high purity human HSCs from cord blood.

In general, the conclusions of this manuscript are substantiated by the experiments the authors described. The sections about flow cytometry-based cell sorting and HSCs transplantations are very convincing. However, the parts on bioinformatics analysis are not always very clear and the readers need to understand how the authors reached their conclusion.

1) The RNA seq experiments described in figure 2 are confusing. The authors state 'our single cell analysis revealed a continuum of cells between those that are enriched in CD90+ CD49f+ (denoted as SP1; Fig. 2a left) and those that are enriched in CD90- CD49f+ (denoted as SP3; Fig. 2a right), with a substantial proportion of cells in-between the two extremities (denoted as SP2) (Fig. 2a and 2b).' However, the data in Fig. 2a are bulk RNA experiments and the populations described are called P1, P2, P4 and P4. Moreover, the CD49f+ populations are called P1 and P2. The single cell data are only showed in Fig. 2b but there are no legend indicating SP1, SP2 and SP3. It's only in Fig. 2c that these names are introduced. In general, the figure 2 is not well described and organised. The authors need to improve it significantly. It will also be useful to explain why they needed bulk RNA seq to identify EPCR as a potential marker. This is not clear.

2) In the sentence 'we verified higher EPCR expression on both CD49f+ compared to the other two CD49f- populations (Supplementary Fig. 2a) and also detected a clear small EPCR+ sub-population (Fig. 2d).' they mentioned Fig. 2d but it does not exist. Did they mean Fig. 3a or 3b?

3) In Figure 5, it is unclear why these specific Wikipathways have been chosen. Besides, the authors did not explain in the methods section how they performed this bioinformatics analysis. More information is needed here.

4) The authors mentioned the concept of 'balanced differentiation'. Could they explain better what they mean by that?

5) In relation to Fig. 6, they used 'gene strength' (page 8) and 'expression strength' (page 21). What do they mean with these terms?

6) On page 9 of the manuscript, they qualify EPCR+ cells as highly homogeneous. They based themselves on the single-cell RNA seq from Fig. 7a. They wrote 'We visualized cellular variability via a t-SNE-based embedding and principal component analysis and this did not reveal evident grouping structure'. However, they did not show any principal component analysis in this figure.

7) The authors used several times the 'lineage-priming modules' terminology. They've taken it from the work by Velten et al. They need to explain in the methods how they defined these lineage-priming nodules in their dataset based on Velten et al. This is not explained currently.

Reviewer #3 (Remarks to the Author):

thanks for the opportunity to review.

This is an excellent and scientifically robust manuscript, detailing the role of CD201 (PROCR/EPCR) in human HSCs identification. The work is based on exploratory analysis using scRNA seq and sequential xeno-transplantation experiments on purified HSC populations.

HSC populations are highly enriched, although not exclusive contained within the EPCR positive CD34+ cells, with multilineage reconstitution capacity. The work reflects previous studies identifying EPCR in stimulated HSCs. The population is transcriptionally homogenous and balanced in function. This work

should provide important information for future studies examining human HSCs. The manuscript does not determine whether EPCR has a functional role or whether it is simply a marker of HSCs, but this is beyond the scope of this report.

Comments:

1. I found the text description of Fig 2 quite confusing, the authors refer to SP1,2,3 which is not actually labelled in the figure, and I can't really appreciate their description of a "continuum of cells". Some clarification/ reordering of the figure would be helpful. Perhaps there would be a better way of showing 2b that is more informative to the overall narrative.

2. Multiple EPCR clones and colours are used. Could you please show the concordance of EPCR staining using the different antibody clones? This will aid reproducibility.

R: We would like to thank all the three reviewers (R1, R2 and R3) for their very positive feed-back, support and pertinent comments. Specific replies can be seen below and we have amended the manuscript with specific-coloured text to each reviewer.

Because R1 requested experiments using MPPs (multipotent progenitors) this resulted in a lot of new data which most of it can be seen in the revised manuscript. Also, we felt that we may have confused R2 regarding the concept of “balance differentiation” and this was mostly related to the in vivo studies (please refer to R2, Q4 for specifics). In order to properly incorporate the transcriptomic data derived from MPPs, we have extended our analysis using more published gene data sets to support the different in vivo lineage outputs observed from the 3 HSPC populations. These gene data sets were derived from transcriptomic analyses using more differentiated cells and at the end the results obtained were very redundant, hence we only used one from Oetjen et al., (ref 9). Based on these three points we had to re-structure Fig. 6 and streamline the data for this figure in order to keep the message concise and effective for all the reviewers and potential readers. We hope all the reviewers can understand this and support these changes.

Reviewer #1 (Remarks to the Author):

The manuscript submitted by Bonnet et al. demonstrated that a specific subpopulation of human HSPCs (CD34+) labelled by EPCR have high repopulating and self-renewal abilities, as well as being transcriptomically and functionally homogenous via single cell analysis. Although the study can provide reasonable evidence, the primary concern is the that features of EPCR in human HSCs have been well characterized in past few years (Fares et al, 2017; Subramaniam et al, 2019), hence the data presented serve as good supporting evidence in another viewpoint, but lacks originality to meet the novelty criteria. More specific comments are listed below.

Major concerns:

1. Here the study defines EPCR+ (vs CD90+EPCR-) populations of human HSCs and proposed a hierarchy of self-renewal capability, this is a nice viewpoint, but doesn't seem to extend the HSC characterization methods significantly, with regard to existing studies and evidence available.

R: we would like to take this opportunity to reply to the generic comment and also to this point altogether. We do recognise the fact that EPCR expression has been described before. We have acknowledged this in the manuscript. As an example, although EPCR has been found to be expressed on human cord-blood expanded cells with UM171 (all the papers published by Prof. Guy Sauvageau's group) but all these works have only demonstrated an association of the percentage of EPCR+ cells in the heterogeneous expanded HSPC population with UM171 with the capacity of having more stem cell activity and the fact that UM171 was able to maintain the presence of EPCR expression in their culture conditions. But none of these works have actually formally demonstrated the existence of a highly purified human HSC population based on EPCR or another surface antigen in unmanipulated cells and most importantly that this same population also exists in the bone marrow.

More importantly, we did/are not claiming the discovery of a novel marker or a new method to identify/enrich human HSCs but rather we have actually identified, studied and proved/provided strong evidences of:

1-the existence of a novel population that contains the highest human HSC frequency (1 in 3 cells) and at the single cell level is the most transcriptomically and functionally homogeneous human HSC population described to date, and this has never been achieved;

2-In addition, we are also showing that not only the widely known CD90+CD49f+ population (defined by Prof. John Dick's group) is enriched in human HSCs but a novel HSC-enriched population exists (CD90-CD49f+). The fact that only these two populations are enriched in human HSCs (and not in other primitive populations) and we were able to identify the true HSCs for the first time within these two populations was based on EPCR expression. It just happens that this highly purified human HSC population expresses EPCR and this surface antigen has been previously described.

3-that the different human primitive HSPC populations described in this study have different lineage output capacities that are supported by different transcriptomic wiring where multipotent EPCR+ HSCs give rise to CD90+EPCR- lymphoid primed progenitors. Therefore, we were able to resolve the priming features of the different HSPC populations that previous works were unable to do (e.g. Velten et al., Nature Cell Biology 2017; PMID: 28319093) because of the poor resolution of the human HSPC hierarchy used that still stands today. Again, the use of CD33 also failed to further enrich human HSCs and thus unable to provide a better understanding of human HSC biology (Knapp et al. Nature Cell Biology 2018; PMID: 29802403).

Overall, our study is able not only to provide a higher resolution of human HSC population but a novel human HSPC hierarchical structure both transcriptionally and functionally. Altogether, these points not only are novel but also offer a very significant advancement to this field and beyond. The fact that EPCR has already been described to be expressed on expanded human HSPCs and on foetal liver cells should not preclude the major impact that this work has to offer to the field of stem cell biology as ultimately it is about the discovery and also the biological characterization of this novel and highly purified human HSC population and not the marker associated with these cells per se.

Maybe we were not clear and did not emphasise enough these messages in our manuscript and we apologise for this. We have re-structured the discussion and now emphasising better the following 7 novel points of this work:

1-The CD90+CD49f+ HSC-enriched fraction does not encompass all human HSCs as we have identified another HSC-enriched population (CD34+CD38-CD45RA-CD90-CD49f+; hereafter CD90-CD49f+) and both populations are interconvertible and give rise to the whole CD34+ HSPCs;

2-Both CD90+CD49f+ and CD90-CD49f+ HSC-enriched populations are still heterogenous and scRNA-sequencing and functional assays demonstrate that only a fraction of cells (~6-10%) within these two populations have long-term repopulating capacity;

3-Within CD90+CD49f+ and CD90-CD49f+, we have identified a subfraction of cells (via a simple cell sorting strategy : CD34+CD38-CD45RA-EPCR+; hereafter EPCR+) which encompasses all CD34+ HSCs;

4-EPCR+ HSCs are multipotent with a unique balanced-myeloid prone primed transcriptomic wiring;

5-EPCR+ HSCs harbour the highest human HSC frequency identified to date, ~1 in 3 cells, which represents a significant improvement from 1 in 10-20 CD90+or-CD49f+ cells thus, closely matching the purities described in mice;

6-EPCR+ HSCs are transcriptomically and functionally very homogeneous at the single cell level and can be easily identified in adult BM;

7-Human primitive HSPCs have a well-structured cell lineage relationship where a novel lymphoid-bias CD90+EPCR+ progenitor cell population, downstream of EPCR+ HSCs, gives rise to MPP cells.

We hope that these arguments will convince this reviewer the impact and importance of this work.

2. Single cell analysis demonstrating that EPCR+ HSCs is homogenous population is reasonable, but it would be much more interesting and informative if single cell analysis was also performed on more heterogeneous population such as CD34+CD38-CD45RA- HSPCs. Even the scRNA-seq result in Figure 2b isn't informative enough to confer distinct signatures among the 2 populations. If some outstanding clusters that specifically express EPCR can be identified, further analysis may reveal more unique signatures associated with EPCR+ clusters.

R: as far we can understand, this reviewer may wish us to perform more single cell analysis with the very heterogeneous CD34+CD38-CD45RA- HSPCs in order to define more unique signatures for the EPCR+ cluster vs other clusters. For this we would need to use 10X genomics or index-sorted HSPCs. Although these methods may allow to investigate the heterogeneity within a population, based on published work this type of assays prove to be ineffective (Velten et al. Nat. Cell Biology 2017) resulted instead at a "CLOUD" (undefined) transcriptomic structure within CD34+CD38-CD45RA- HSPCs. Many reasons could explain this: 1- scRNA-seq in very quiescent/low metabolic cells, like human HSPCs, is challenging (due to low quantity of RNA per cell) and this can cause high levels of noise; 2-continuum transcriptomic nature of human HSPCs as demonstrated by Velten et al. The latter may be the strongest reason why it was not even possible to achieve clustering of CD90+CD49f- cells and MPPs (CD90-CD49f-) in this study and in other subsequent scRNA-seq studies by other groups (Pellini et al., Nat. Comm. 2019 PMID: 31160568; Ranzoni et al., Cell Stem Cell 2021 PMID: 33352111). These were the concerns we had, hence we decided not to perform such expensive experiments in the first place as it was likely to fail to define any clustering within CD34+CD38-CD45RA- HSPCs. Although we could attempt to perform such experiments now but unfortunately currently our sequencing facilities (both at the CRICK and in Cardiff University) have a long backlog of work as most staff have been re-allocated to perform internal COVID19-testing. As such, we have some logistical and timing issues on how we can perform many pre-testing/optimizations first (that may be in vain) before the final experiments. I hope this reviewer can understand the current situation. Having said that, fortunately we did perform RNA-seq in MPPs and since this reviewer is also requesting to add more data using MPPs (point 3) we are happy to include the transcriptomic data derived from this population.

We re-analyzed/compared MPPs with EPCR+ vs CD90+EPCR- (the 3 main population within CD34+CD38-CD45RA- HSPCs) and now we are proving a more detailed transcriptomic signatures of EPCR+ HSCs at the population level. Despite our effort, no major new transcriptomic signatures were uncovered due to the very close relationship between these 3 primitive populations, but it did reinforce our initial analysis and we would like to thank this reviewer for requesting this data.

3. Most assays were performed on EPCR+ vs CD90+EPCR- populations, it would be more comprehensive if CD90-EPCR- population identified as MPPs were also included in the analysis (Figure 4,5, 6).

R: Although MPPs have already been extensively studied/described by others (e.g. Notta et al., Science 2016; Karamitros et al., Nat. Immunology 2018) but we can appreciate the rational of this reviewer by requesting more analysis on MPPs. Since we are describing a new human HSPC hierarchical structure and two novel populations, EPCR+ HSC and its immediate downstream population CD90+EPCR- (both are above MPPs), hence MPPs can serve as a reference point to other published works.

Since the newly defined MPP population (EPCR-CD90- (CD34+CD38-CD45RA-)) is very similar (if not the same) to CD49f-CD90- (CD34+CD38-CD45RA) MPPs, the in vivo data using CD49f-CD90- MPPs is already shown in Figures 1- to 3. Therefore, we did not repeat the in vivo LDA and kinetics experiments

as it would be redundant. However, we did perform some in vivo experiments using the newly defined MPPs that are now shown in the new Fig. 4c, Fig. 6a and Supplementary Fig. 3a. Both new Figs. 5 and 6 have now the MPP-derived data throughout (transcriptomic and functional data). Once again, we thank this reviewer, as with these new results it made our manuscript stronger.

Minor concerns

1. How well is CD49f expression correlating to EPCR expression in CD34+ cells? Authors may consider to plot CD49f vs EPCR to examine the patterns.

R: this was actually partially shown in Fig. 2a where selected CD34+CD38-CD45RA-EPCR+ cells were plotted against CD49f stains (bottom facs plot) and also shown in the old Supplementary Fig. 2b (now as Supplementary Fig. 2c). We also provided the opposite situation where the four CD34+CD38-CD45RA-CD90+or-CD49f+or- populations were plotted against EPCR stains (Fig. 3a). Nevertheless, as requested, we are providing the CD49f vs EPCR plot in CD34+CD38-CD45RA- cells (now as Supplementary Fig. 2b).

2. Have the authors tried t-SNE or UMAP representation to visualize in Figure 2b?

R: We chose to visualize the single-cell data using a diffusion map approach, since this algorithm is designed to capture smooth transitions between cells. In contrast, t-SNE and UMAP are better suited to capture distinct cell clusters. In fact, as illustrated below, the t-SNE representation is able to illustrate some degree of cell continuum between the two single CD49f+ cells, but this continuum is less evident as compared to the one identified by the diffusion map algorithm. We added a justification for using diffusion map algorithm in the manuscript to clarify this point.

3. Figure 2d that is described in the manuscript went missing in the actual figure.

R: apologies. We amended to Fig. 3a.

4. Figure 5, it will be more informative if the author could perform a few additional functional assays to validate EPCR+ vs CD90+EPCR- HSC features based on GO analysis. Functional assay that may be important for validation includes OP-puro protein synthesis (ribosomal proteins), seahorse XF or cell tilter glo (OXPHOS).

As requested, we performed Cell-TitreGlo (for ATP) and also OP-Puro incorporation assay. All is shown in the new Fig. 5.

Reviewer #2 (Remarks to the Author):

Comments to the authors:

Anjos-Afonso et al. present in this manuscript a new method to isolate with high purity human hematopoietic stem cells (HSCs). This is a very important contribution to the field. The transplantation of HSCs is a life saving procedure for many patients across the world. Considering that sources of HSCs are limited and therefore very precious, being able to isolate HSCs with the highest purity would ensure that they are best used in medical applications. Here the authors used flow cytometry to characterize and isolate HSCs and combine it to single-cell transcriptomics to identify new cell surface markers characteristic of human HSCs. This is how they found the marker EPCR which allowed them to isolate with high purity human HSCs from cord blood.

In general, the conclusions of this manuscript are substantiated by the experiments the authors described. The sections about flow cytometry-based cell sorting and HSCs transplantations are very convincing. However, the parts on bioinformatics analysis are not always very clear and the readers need to understand how the authors reached their conclusion.

1) The RNA seq experiments described in figure 2 are confusing. The authors state 'our single cell analysis revealed a continuum of cells between those that are enriched in CD90+ CD49f+ (denoted as SP1; Fig. 2a left) and those that are enriched in CD90- CD49f+ (denoted as SP3; Fig. 2a right), with a substantial proportion of cells in-between the two extremities (denoted as SP2) (Fig. 2a and 2b).' However, the data in Fig. 2a are bulk RNA experiments and the populations described are called P1, P2, P4 and P4. Moreover, the CD49f+ populations are called P1 and P2. The single cell data are only showed in Fig. 2b but there are no legend indicating SP1, SP2 and SP3. It's only in Fig. 2c that these names are introduced. In general, the figure 2 is not well described and organised. The authors need to improve it significantly. It will also be useful to explain why they needed bulk RNA seq to identify EPCR as a potential marker. This is not clear.

R: our apologies. This reviewer is absolutely correct, and we have revised this part of the manuscript: 1-we changed/added some extra legends and re-organised the figure.

2-we are also providing an extensive explanation in the manuscript (page 5 and page 31-32 (M&M section); of note, the bulk RNA seq data was used in two different ways: first, as both CD90+CD49f+ and CD90-CD49f+ single cells have a continuum transcriptomic nature, hence it was difficult to define the common population between them. Thus, we used the DE list C that was obtained by comparing the DE of CD90+CD49f+ (P1) vs the other two CD49f- populations (P3+P4) + the DE of CD90-CD49f+ (P2) vs the other two CD49f- populations (P3+P4) to untangle this issue (Fig. 2a and 2c). In addition, the bulk RNA seq also served as supporting or confirmation data that we were able to identify EPCR independently from the scRNA- seq analysis.

2) In the sentence 'we verified higher EPCR expression on both CD49f+ compared to the other two CD49f- populations (Supplementary Fig. 2a) and also detected a clear small EPCR+ sub-population (Fig. 2d).' they mentioned Fig. 2d but it does not exist. Did they mean Fig. 3a or 3b?

R: apologies, that was incorrect. We amended to Fig. 3a.

3) In Figure 5, it is unclear why these specific Wikipathways have been chosen. Besides, the authors did not explain in the methods section how they performed this bioinformatics analysis. More information is needed here.

R: The set of pathways shown in Fig. 5 is the subset of significantly enriched pathways based on the entire WikiPathways database. To identify these pathways, we performed pathway analysis using camera, a competitive gene set test accounting for inter-gene correlation (ref. 15). Pathway

enrichment was considered significant at an FDR of 0.05. We added this information, and details on this analysis are now provided in the Materials and Methods in the “Bulk population RNA-sequencing” section (now page 31).

4) The authors mentioned the concept of ‘balanced differentiation’. Could they explain better what they mean by that?

R: the notion of a “balance differentiation” output from EPCR+ HSCs that we wanted/want to convey was/is mainly related to the in vivo differentiation capacity of EPCR+ HSCs vs the other two progenitor populations (CD90+EPCR- and now including data on MPP). As an example, the well documented in vivo output in immunodeficient mice is normally CD19+ bias (~80-90% of human cells are CD19+). This was no different when we transplanted CD90+EPCR- cells and MPPs (revised Fig. 6a) however, EPCR+ HSCs gave a more equal CD19+ vs CD33+ outputs as compared to other tested populations, even being more myeloid prone. We are now providing more explanations throughout the manuscript and in few instances, we also changed “balanced” to “multilineage” as is more appropriate.

Of note, we have previously shown some in vitro differentiation data using EPCR+ vs CD90+EPCR- cells on MS5 “to support” the in vivo balance differentiation output notion. We decided to eliminate this comparison in the revised manuscript, as NSG mouse model provides a B cell-bias output whereas MS5 provides more a myeloid output. In addition, CD56+ NK cells don’t develop well in NSG mice, hence overall it was not a good supporting data and it was/can be quite confusing. We only kept parts of this data (from EPCR+ cells only) mainly to convey the message that these HSCs are multilineage at the population level and also at the single cell level (to link the data shown in Fig. 7; and this can be seen in the new Supplementary Fig. 6a) but omitted any comparison with other populations and we are no longer using this data to support the in vivo balanced differentiation point of view. In hindsight, we did ourselves a disservice and confused this reviewer. Our apologies and we thank this reviewer that allowed us to refine this message.

5) In relation to Fig. 6, they used ‘gene strength’ (page 8) and ‘expression strength’ (page 21). What do they mean with these terms?

R: Our apologies. We wanted to mean “Gene Expression Strength”. We have amended throughout the manuscript. The gene expression strength is the overall weight expression of the genes comprised in each gene set or module. This was done by controlling for cell type (in this case EPCR+ HSCs) when regressing out the gene expression data, in other words when regressing the normalized counts of the genes in each module it was taking in consideration the overall gene expression found in the testing cell population. We added this information in the M&M (page 31) and in the main text (page 11).

6) On page 9 of the manuscript, they qualify EPCR+ cells as highly homogeneous. They based themselves on the single-cell RNA seq from Fig. 7a. They wrote ‘We visualized cellular variability via a t-SNE-based embedding and principal component analysis and this did not reveal evident grouping structure’. However, they did not show any principal component analysis in this figure.

R: apologies, we added this data now in the revised Supplementary Fig.5 (left plot).

7) The authors used several times the ‘lineage-priming modules’ terminology. They’ve taken it from the work by Velten et al. They need to explain in the methods how they defined these lineage-priming nodules in their dataset based on Velten et al. This is not explained currently.

R: The lineage-priming modules based on Velten et al and now on Oetjen et al. (derived from more differentiated cells) studies were actually determined/defined by the authors of their respective works. The list of modules (and the associated genes) shown in our Supplementary Table 4 were taken

(downloaded) directly from the supplementary data of the respective publications. However, we did organize their data in a more coherent manner as most of the data is scattered in their main data and supplementary information. We basically used our DE gene lists between the 3 HSPCs and performed NES (using GSEA platform), gene expression strength, etc analyses using Velten et al's and Oetjen et al's gene modules as our gene set database.

We added this information in the materials and methods (end of page31). We hope that we were able to clarify this point but we are happy to provide further clarification if this reviewer is still unclear on how we did our analysis. Thank you.

Reviewer #3 (Remarks to the Author):

thanks for the opportunity to review. This is an excellent and scientifically robust manuscript, detailing the role of CD201 (PROCR/EPCR) in human HSCs identification. The work is based on exploratory analysis using scRNA seq and sequential xeno-transplantation experiments on purified HSC populations. HSC populations are highly enriched, although not exclusive contained within the EPCR positive CD34+ cells, with multilineage reconstitution capacity. The work reflects previous studies identifying EPCR in stimulated HSCs. The population is transcriptionally homogenous and balanced in function. This work should provide important information for future studies examining human HSCs. The manuscript does not determine whether EPCR has a functional role or whether it is simply a marker of HSCs, but this is beyond the scope of this report.

Comments:

1. I found the text description of Fig 2 quite confusing, the authors refer to SP1,2,3 which is not actually labelled in the figure, and I can't really appreciate their description of a "continuum of cells". Some clarification/ reordering of the figure would be helpful. Perhaps there would be a better way of showing 2b that is more informative to the overall narrative.

R: our apologies. Similar question was raised by R2; we have extensively revised this part (please refer to point 1 from R2 and pages 5 and 30-31 of the manuscript).

2. Multiple EPCR clones and colours are used. Could you please show the concordance of EPCR staining using the different antibody clones? This will aid reproducibility.

R: all 3 antibody clones used (when conjugated with APC) resulted in a similar staining profile and revealed that almost all CD34+CD38-CD45RA-EPCR+ cells were CD49f+ as shown now in the Supplementary Fig. 2c. Studies that required CD49f (with anti-CD49f PE), TMRE stains (Fig. 5d) and other PE conjugated antibodies stains (Supplementary Fig.6), the CD201 APC clone RCR-227 antibody was used, with the exception in the experiments just to illustrate the data shown in Supplementary Fig. 2c. We started/continued using the clone RCR-227 (APC conjugated) in some in vivo studies when CD49f stain was no longer required. However, since we observed a better separation of the EPCR+ vs EPCR- populations when PE conjugated antibodies were used, as PE is a brighter fluorochrome, we subsequently changed to the CD201 PE clone RCR-401 antibody (simply because a CD201 PE clone RCR-227 antibody does not commercially exist). Of note, we observed no differences in the in vivo engraftment when cells were isolated using one or the other antibody (see "Supplementary Table 1 - in vivo LDA.xlsx" file spreadsheet "EPCR+ HSC vs CD90+EPCR-"). All the information is now added in the revised manuscript (in the figure legends) as requested in particular in the "Supplementary Table 1 - in vivo LDA.xlsx" file spreadsheets "CD90+or-CD49f+EPCR+" and "EPCR+ HSC vs CD90+EPCR-".

REVIEWERS' COMMENTS

Reviewer #1 (Remarks to the Author):

None

Reviewer #2 (Remarks to the Author):

My comments were addressed satisfactorily.

Reviewer #3 (Remarks to the Author):

Thank you. The authors have addressed my concerns from the first round of review. These are important observations and will add incrementally to resources/understanding of human HSC and hematopoiesis.